# Algorithmic Stability Based Generalization Bounds for Adversarial Training

**Runzhi Tian**
University of Ottawa
rtian081@uottawa.ca

**Yongyi Mao**
University of Ottawa
ymao@uottawa.ca

## Abstract

In this paper, we present a novel stability analysis of adversarial training and prove generalization upper bounds in terms of an expansiveness property of adversarial perturbations used during training and used for evaluation. These expansiveness parameters appear to not only govern the vanishing rate of the generalization error but also govern its scaling constant. Our bound attributes the robust overfitting in PGD-based adversarial training to the sign function used in the PGD attack, resulting in a bad expansiveness parameter. The peculiar choice of sign function in the PGD attack appears to impact adversarial training both in terms of (inner) optimization and in terms of generalization, as shown in this work. This aspect has been largely overlooked to date. Going beyond the sign-function based PGD attacks, we further show that poor expansiveness properties exist in a wide family of PGD-like iterative attack algorithms, which may highlight an intrinsic difficulty in adversarial training. Code is available at https://github.com/rzTian/AT-Stability.

## 1 Introduction

Deep neural networks, despite their great success, have been shown vulnerable to adversarial attacks (Szegedy et al., 2014; Goodfellow et al., 2015), where carefully constructed small modifications of the input may cause the network to output a wrong prediction. A large body of works (Madry et al., 2019; Zhang et al., 2019; Croce et al., 2020; Shaham et al., 2018; Qin et al., 2019; Shafahi et al., 2019; Wong et al., 2020) then propose revised training algorithms to combat adversarial attacks. These algorithms, usually referred to as adversarial training (or AT in this paper), among which the dominant approaches, such as PGD based AT (Madry et al., 2019), involve perturbing the input in a way similar to adversarial attacks to hopefully maximize the loss function within a prescribed radius (referred to as "inner maximization"). Although these AT algorithms allow the learned model to defend, to some extent, against adversarial attacks, significant challenges remain.

First, generalization for such training algorithms is much more difficult, a phenomenon known as "robust overfitting"(Rice et al., 2020). Specifically, Rice et al. (2020) shows that on the CIFAR-10 dataset (Krizhevsky et al., 2009), the model trained by AT using 10-step PGD attack is still vulnerable to the same 10-step PGD attack on the testing set. Our additional experi-

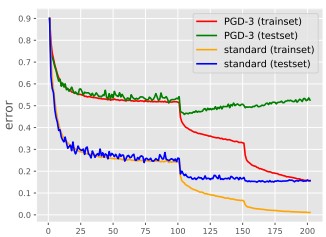

Figure 1: The learning curve of a model trained by AT on CIFAR-10 with 3-step PGD. The standard error as well as the error against the same 3-step PGD attack are measured during AT on both the training and testing sets. The step size for PGD and the perturbation radius w.r.t the $\infty$−norm are respectively set to $7/255$ and $8/255$. The learning rate is decayed at the $100^{\text{th}}$ and the $150^{\text{th}}$ epoch.

ments (e.g., Figure 1) suggests that this is quite common. In Figure 1, we perform AT with a 3-step PGD and measure the error of the model against 3-step PGD attack as well as its standard error in the training process. We observe that the model trained with 3-step PGD is still vulnerable to the same PGD attack on the testing set. After the first learning rate decay (the $100^{\text{th}}$ epoch), the testing error w.r.t the 3-step PGD starts to rise, similar to the observations in Rice et al. (2020).

Second, it is much more difficult to develop theoretical understanding of the generalization behavior for models obtained from AT, comparing with those from standard training. In that direction, some theoretical works consider the setting where the inner maximization is perfectly solved, e.g., in Yin et al. (2018); Awasthi et al. (2020). However, such settings are invalid for more complex neural networks, where the closed-form solution for the inner maximization is unavailable. Another line of works use uniform stability to analyze the generalization of AT. In Xing et al. (2021), the adversarial loss is assumed convex and non-smooth and AT is regarded as standard SGD on this loss, whereby an existing generic bound for non-smooth loss in Bassily et al. (2020) is invoked for analysis. As pointed out in Xiao et al. (2022b), the bound obtained in Xing et al. (2021) is independent of the specific choice of loss function used for training and insufficient to reflect the difference between AT and SGD observed in practice. The work of Xiao et al. (2022b) argues that the adversarial loss is approximately smooth and derive bounds based on the stability framework of SGD in Hardt et al. (2016). The work of Wang et al. (2024), built upon Xiao et al. (2022b), extends the analysis of AT to the data-dependent stability framework in Kuzborskij & Lampert (2018). But the bounds obtained in both Xiao et al. (2022b) and Wang et al. (2024) do not vanish with sample size.

To overcome these limitations and shed new light in understanding robust overfitting, we present in this work novel stability analysis for the generalization of models learned using an arbitrary AT algorithm. Specifically, we isolate two aspects in the problem scope. The first is the loss function used for performance evaluation (on both testing and training sets), which can in general be considered as a modification of the standard loss to a version induced by a perturbation map $J$ and which reduces to the adversarial loss when $J$ is specialized to a particular form $J^*$ and to the standard loss when $J$ is specialized to the trivial identity map $J^{\text{id}}$. The second is the perturbation map $\pi$ used during AT, corresponding to the solution heuristics used for solving the inner maximization problem. When allowing $\pi$ to potentially deviate from the perturbation $J^*$, we include in our study the case where the inner maximization is not solved exactly. Additionally, considering $\pi = J$ allows us to study robust overfitting as examplified in Figure 1 where training using a particular attack results in poor generalization when evaluation on the testing set is under the *same* attack.

In this setting, we carry out a stability analysis and present novel generalization bounds for models trained using AT with an arbitrary adversarial perturbation $\pi$ and evaluated on a loss induced by an arbitrary perturbation $J$. At the heart of our analysis is the introduction of a notion of "expansiveness" for the perturbation maps ($J$ and $\pi$), which governs the behavior of the derived bounds. Specifically, we show that whenever the expansiveness parameter of $J$ is strictly bounded, our generalization bounds vanish with sample size $n$ as $\mathcal{O}(1/n)$ and a small expansiveness parameter of $\pi$ further helps generalization. On the other hand, when the $J$-loss (i.e., the loss induced by perturbation $J$) is defined with $J$ taken as the sign-PGD perturbation, the expansiveness parameter of $J$ is no longer bounded. In this case, our bound reveals an intrinsic tension between the stability parameter, and the perturbation radius, and the ambient data dimension, in their respective roles on generalization – specifically, the bound converges to a constant. Additional advantages of our bounds include the following. Our generic bound (Theorem 4.2) is applicable to AT algorithms based on any form of adversarial perturbations. Our bounds do not rely on any assumption on the adversarial loss directly, since we only make assumptions on the standard loss and all properties of the adversarial loss are induced via perturbation map $J$. Finally, varying the form of $J$ potentially enables this framework to be applicable to settings where generalization on other performance metrics is of interest.

We zoom into models trained with multi-step PGD, and further demonstrate that the sign function used in the perturbation is an important cause of robust overfitting for such AT methods. We experimentally replace the sign function in PGD with a smooth approximation $\tanh_\gamma$, where $\tanh_\gamma(x) = \tanh(\gamma x)$ and the parameter $\gamma$ controls the smoothness of the function and hence the expansiveness of the PGD perturbation (decreasing $\gamma$ decreases the expansiveness). Our experiments show that reducing $\gamma$ results in smaller generalization gaps. These results validate our bound and its implication on generalization. Interestingly our experiments also reveal that sign-PGD appears as a stronger attack than $\tanh_\gamma$-PGD and the raw gradient (RG)-PGD attack, even on the training set. Performing AT with $\tanh_\gamma$-PGD and RG-PGD may be inadequate for defending against the sign-PGD attacks on the training set. Our observations suggest that sign-function, a building block of PGD-based AT, appears to play a peculiar role: comparing with the $\tanh_\gamma$ counter-part, the sign function helps to better solve the inner maximization problem but at the same time cause the perturbation $\pi$ to suffer from bad expansiveness and results in poor generalization. This aspect of

sign-PGD has been largely overlooked to date, since most theoretical analysis of PGD removes the sign function in their consideration (i.e., studying RG-PGD instead).

In this work, we also recognize sign-PGD as an iterative method for solving the inner maximization problem where each step is principled by a locally linear approximation of the loss function. Based on this principle, we extend sign-PGD to a wider family of perturbations. We show theoretically that every member in this family suffers from poor expansiveness. This result seems to point to certain intrinsic difficulty in training models adversarially.

## 2 OTHER RELATED WORKS

**Robust generalization** Beyond investigations via algorithmic stability perspectives (e.g., Xing et al. (2021); Xiao et al. (2022b); Wang et al. (2024)), robust generalization has also been studied under the uniform convergence framework with conventional statistical learning tools such as VC dimension (Montasser et al., 2019), Rademacher complexity (Khim & Loh, 2019; Yin et al., 2018; Awasthi et al., 2020; Xiao et al., 2022a; Attias et al., 2018) and other PAC learning frameworks (Cullina et al., 2018; Diochnos et al., 2019). Moreover, robust generalization has been investigated via the curvature of the local minima of the loss landscape: AT is observed to have a tendency to reach sharper minima (Liu et al., 2020), and flatter minima usually results in better generalization (Wu et al., 2020). The work of Chen et al. (2020) observes that robust overfitting can be alleviated by smoothing the model prediction via knowledge distillation. The difficulty of achieving robust generalization has also been attributed to the inadequate expressive power of practical deep learning models (Li et al., 2022), insufficient sample size for models to generalize (Schmidt et al., 2018) as well as and the model's tendency to interpolate "hard training instances" (Liu et al., 2021).

**Uniform stability** Uniform stability was first introduced by the landmark work of Bousquet & Elisseeff (2002). An influential work by Hardt et al. (2016) adapts this framework to analyze the uniform stability of SGD with smooth loss functions, explaining the effectiveness of SGD in training neural networks. Since then, many studies have built upon Hardt et al. (2016) to develop stability bounds for SGD with non-smooth losses (e.g., Bassily et al. (2020); Lei & Ying (2020)). Data-dependency in stability analysis is introduced in Kuzborskij & Lampert (2018), and uniform stability for more sophisticated variants of SGD is also studied (e.g., Mou et al. (2018); Chen et al. (2018)). Additionally, works such as Farnia & Ozdaglar (2021); Lei et al. (2021) have explored algorithmic stability in general minimax problems. These studies are more closely related to generative adversarial networks (GANs), rather distant from the standard settings of adversarial training.

## 3 PROBLEM SETUP AND PRELIMINARIES

Over any real vector space, we will use $\| \cdot \|_p$ to denote the $p$-norm and abbreviate the Euclidean norm (i.e., 2-norm) as $\| \cdot \|$. For a vector $x \in \mathbb{R}^d$, $x[i]$ denotes the $i^{\text{th}}$ coordinate of $x$.

We consider the standard setting of supervised learning, where the training samples are instance-label pairs, $(x_i, y_i)$'s, drawn i.i.d from an underlying data distribution $\mathcal{D}$ over $\mathcal{X} \times \mathcal{Y}$. Here the input space $\mathcal{X}$ is $\mathbb{R}^d$ and the label space $\mathcal{Y}$ is finite. We restrict to parameterized models, e.g., neural networks, in which the model parameter $w$ lives in a subset $\mathcal{W}$ of some real vector space. We use $f(w, x, y)$ to denote the loss value of $(x, y)$ under model parameter $w$, where a standard choice of loss function (e.g. 0-1 loss, cross-entropy loss, etc.) is absorbed in $f$. For example, $f(w, x, y)$ can be the cross-entropy loss of the a neural network with parameter $w$ on sample $(x, y)$.

The central object of this study is adversarial training, which allows the learned model to resist adversarial attacks. Each adversarial attack (or adversarial perturbation) on input $x$ is assumed to live in an $\infty$-norm ball $\mathbb{B}_\infty(x, \epsilon) := \{t \in \mathbb{R}^d : \|t - x\|_\infty \leq \epsilon\}$ with radius $\epsilon$ and centered at $x$.

**Perturbation induced loss** Let $J$ be a function mapping $\mathcal{W} \times \mathcal{X} \times \mathcal{Y}$ to $\mathcal{X}$ satisfying $J(x; y, w) \in \mathbb{B}_\infty(x, \epsilon)$. Then $J(x; y, w)$ may be regarded as a perturbation of $x$ by a magnitude of up to $\epsilon$ (under $\infty$-norm). We then define the *perturbation J induced loss* or simply *J-loss* by

$$f_J(w, x, y) := f\left(w, J(x; y, w), y\right) \tag{1}$$

Let $J^*(x; y, w) := \arg\max_{\hat{x} \in \mathbb{B}(x,\epsilon)} f(w, \hat{x}, y)$, and $J^{\mathrm{id}}(x; y, w) := x$. Then it is easy to verify that when $J = J^*$, $f_J(w, x, y)$ is the adversarial loss $\max_{\hat{x} \in \mathbb{B}(x,\epsilon)} f(w; \hat{x}, y)$ —for which reason, we will denote the adversarial loss $f_{J^*}$ by $f^*$ for simplicity —and when $J = J^{\mathrm{id}}$, $f_J(w, x, y)$ is the standard loss $f(w, x, y)$. We will soon encounter other forms of $J$-loss.

**Generalization w.r.t the induced loss**  Let the training set $S = \{(x_i, y_i)\}_{i=1}^{n}$ be drawn from $\mathcal{D}^n$. Consider a learning algorithm $A$, which when applied on $S$ gives rise to a learned model parameter $w = A(S)$. Notably $w$ entails randomness, due to the random sampling of $S$ and the possible intrinsic randomness in $A$. The population risk and empirical risk w.r.t $J$-loss are defined respectively as:

$$R_{\mathcal{D}}[A(S); J] := \mathbb{E}_{(x,y) \sim \mathcal{D}} [f_J(A(S), x, y)] \quad \text{and} \quad R_S[A(S); J] := \frac{1}{n} \sum_{i=1}^{n} f_J(A(S), x_i, y_i)$$

Note that $R_{\mathcal{D}}[A(S); J]$ and $R_S[A(S); J]$ are both random variables. The expected generalization gap w.r.t the $J$-loss is then

$$\mathrm{GG}_n(J, A) := \mathbb{E}_{S,A} [R_{\mathcal{D}}[A(S); J] - R_S[A(S); J]]$$

where expectation over $A$ refers to averaging over the intrinsic randomness in $A$. Specially, we will call $\mathrm{GG}_n(J^{\mathrm{id}}, A)$ and $\mathrm{GG}_n(J^*, A)$ respectively the *standard generalization* gap and the *robust generalization* gap of the algorithm $A$.

The generalization gap can be analyzed by exploiting the tool of uniform stability (Bousquet & Elisseeff, 2002). We say that the algorithm $A$ is $\rho-$uniformly stable w.r.t $J$-loss, if

$$\Delta_n(J, A) := \sup_{S \simeq S'} \sup_{(x,y) \in \mathcal{X} \times \mathcal{Y}} \mathbb{E}_A[f_J(A(S), x, y) - f_J(A(S'), x, y)] \leq \rho \tag{2}$$

Here $S \simeq S'$ denotes two datasets that each contains $n$ samples but differ in at most one. It is shown in Hardt et al. (2016) that uniform stability implies generalization in expectation, namely,

**Lemma 3.1** (Hardt et al. (2016)). *For any perturbation $J$ and any algorithm $A$,*

$$\mathrm{GG}_n(J, A) \leq \Delta_n(J, A) \tag{3}$$

The lemma is due to that the analysis Hardt et al. (2016) applies to arbitrary loss functions, including the $J-$loss defined above. In our work, we will consider the family of $f$ that are Lipschitz and gradient-Lipschitz with respect to both $x$ and $w$ in the following sense: there exist positive constants $L_{\mathcal{X}}, L_{\mathcal{W}}, \Gamma_{\mathcal{X}}$ and $\beta$ such that for any $y \in \mathcal{Y}$, any $x, x' \in \mathcal{X}$ and any $w, w' \in \mathcal{W}$

$$|f(w', x', y) - f(w, x, y)| \leq L_{\mathcal{X}} \|x - x'\| + L_{\mathcal{W}} \|w - w'\| \tag{4}$$

$$\|\nabla_{w'} f(w', x', y) - \nabla_w f(w, x, y)\| \leq \Gamma_{\mathcal{X}} \|x - x'\| + \beta \|w - w'\| \tag{5}$$

Similar Lipschitzness and smoothness assumptions are also used in other stability analysis literature, as in Hardt et al. (2016); Farnia & Ozdaglar (2021); Xiao et al. (2022b); Wang et al. (2024).

With the Lipschitz condition of $f$, the uniform stability w.r.t $f_J$ can be related to the notion of the uniform argument stability (UAS), a notion coined in Bassily et al. (2020), as well as an "expansiveness" property of $J$, which we will soon define. Specifically the UAS parameter of $A$ is

$$\delta_n(A) := \sup_{S \simeq S'} \mathbb{E}_A \|A(S) - A(S')\|$$

and for any given $c \geq 0$, we define the $c$-expansiveness of perturbation $J$ as

$$q_c(J) := \sup_{(x,y)} \sup_{w, w': \|w - w'\| > c} \frac{\|J(x; y, w) - J(x; y, w')\|}{\|w - w'\|}$$

We note that such a notion of expansiveness reduces to a Lipschitz condition when $c = 0$. It measures the sensitivity of an operator to the perturbation of its input, sharing similarity with the Lipschitz condition but provide extra benefit when analyzing operators whose Lipschitz constant is unbounded. When taking $c > 0$, this expansiveness, however, excludes measuring sensitivity for perturbation with magnitude lower than $c$. This consideration is motivated by the fact that in

practice, extremely small perturbation do not arise. Additionally, this expansiveness behaves nicely, i.e., being bounded, even for non-continuous operators, such as those defined via the sign function, to arise later in this paper.

For any given $S$ and $S'$ differing by only one element and every $c^* \geq 0$, let $Q(S, S'; c^*)$ denote the probability (under the probability measure induced by the randomness in $A$) that $\|A(S) - A(S')\| < c^*$. Specifically, let $S_*$ and $S'_*$ denote two training sets with $S_* \simeq S'_*$ which achieve the supremum in the definition of $\Delta_n(J, A)$ in (2). We write $Q(c^*)$ in place of $Q(S_*, S'_*; c^*)$ for simplicity.

**Lemma 3.2.** *If the loss function $f$ satisfies the Lipschitz condition (4), then for any $c^* \geq 0$,*

$$\Delta_n(J, A) \leq (L_{\mathcal{W}} + q_{c^*}(J)L_{\mathcal{X}})\delta_n(A) + L_{\mathcal{X}}Q(c^*) \cdot 2\epsilon\sqrt{d} \tag{6}$$

The proof of this lemma is deferred to Appendix A. In the remainder of this paper, we will use this bound to analyze the generalization of adversarial training (AT) algorithms. We will show, for most cases, that this bound vanishes with sample size $n$ by choosing a judicious choice of $c^*$. The only case in which a vanishing bound is not attainable is sign-PGD based AT, where the bound converges to a constant. This may reveal some intrinsic difficulty in generalization for such AT algorithm.

## 4 UNIFORM STABILITY OF ADVERSARIAL TRAINING

Lemma 3.2 suggests that the expansiveness of the perturbation $J$, which is used to define the $J$-loss $f_J$, plays a role in generalization. We now take $A$ as an *adversarial training (AT) algorithm* where we will show that the expansiveness of the perturbation used in the AT training algorithm $A$ plays another role by impacting the UAS parameter $\delta_n(A)$.

**AT algorithms** We consider the following iterative AT algorithm. At each iteration of AT, it first draws a training sample $(x_{i_t}, y_{i_t}) \in S$ and then updates the model parameter $w_t$ according to

$$x_t^{\text{adv}} = \pi(x_{i_t}; y_{i_t}, w_t) \tag{7}$$

$$w_{t+1} = w_t - \tau_t \nabla_{w_t} f(w_t, x_t^{\text{adv}}, y_{i_t}) \tag{8}$$

Here $\tau_t \in \mathbb{R}_+$ denotes the step size of the gradient descend at the iteration $t$, $i_t \in \{1, \cdots, n\}$ is drawn uniformly and independently (across $t$) from $\{1, 2, \ldots, n\}$, and $\pi(x_{i_t}; y_{i_t}, w_t)$ denotes perturbation of $x_{i_t}$ within $\mathbb{B}_\infty(x_{i_t}, \epsilon)$. We note that ideally $\pi$ should be $J^*(x_{i_t}; y_{i_t}, w_t)$ but in practice it is only an approximation of it due to the difficulty in acquiring the exact solution. Additionally and more critically, we note that, despite that both $\pi$ and $J$ refer to perturbations, the two notions in this paper may be completely different. Specifically, $J$ induces the $J$-loss, which is used as a performance metric (evaluated either on the training set or on the testing set), whereas $\pi$ denotes the perturbation operation applied during adversarial training. Although in some cases $\pi$ is $J$ or is related to $J$, there are scenarios in which $\pi$ and $J$ are completely decoupled, for example, when we perform adversarial training but choose to evaluate the model using the standard loss, i.e., using $J^{\text{id}}$-loss. In a later section, we will see more cases in which $J$ and $\pi$ are completely different. As a minor comment, we note that when the perturbation $\pi$ in (7) is chosen as the identity map $J^{\text{id}}$, the AT algorithm reduces to the standard stochastic gradient descend (SGD) algorithm. Finally, as we may look into various choices of $\pi$ in AT algorithms, we use $A_\pi$ to denote an AT algorithm, emphasizing its dependence on $\pi$. Under such notations, we may even consider "mis-matched generalization gap", namely, $\text{GG}_n(J, A_\pi)$ with $J \neq \pi$, for example, $J = J^{\text{id}}$ and $\pi$ is a particular adversarial perturbation.

Note that although $x_t^{\text{adv}}$ is also a function of $w_t$, the derivative operator in (8) does not go through $\pi$, an option consistent with the standard AT implementation as in Madry et al. (2019); Rice et al. (2020).

We now present an upper bound for the UAS of AT.

**Theorem 4.1.** *Suppose that $f$ satisfies the conditions (4) and (5). If we run $A_\pi$ for $T$ steps with step sizes $\tau_t \leq \frac{1}{\beta}$, there exists a constant $c > 0$ such that we have*

$$\delta_n(A_\pi) \leq \frac{2L_{\mathcal{W}}}{n\beta} \sum_{t=0}^{T} (2 + q_c(\pi)\Gamma_{\mathcal{X}}/\beta)^t \tag{9}$$

We defer the proof of the theorem to Appendix A. With the upper bound of the UAS, an upper bound for the mismatched generalization gap can be immediately derived according to (3) and (6) as below:

**Theorem 4.2.** *Under the condition of Theorem 4.1, for any $c^* \geq 0$, there exists a constant $c > 0$, such that*

$$\mathrm{GG}_n(J, A_\pi) \leq (L_{\mathcal{X}} q_{c^*}(J) + L_{\mathcal{W}}) \frac{2L_{\mathcal{W}}}{n\beta} \sum_{t=0}^{T} (2 + q_c(\pi)\Gamma_{\mathcal{X}}/\beta)^t + L_{\mathcal{X}} Q(c^*) \cdot 2\epsilon\sqrt{d} \tag{10}$$

The bound in (10) also includes as a special case the "matched" generalization gap $\mathrm{GG}_n(J, A_J)$, where the perturbation used in adversarial training is identical to that defining performance metric, as is typical in the adversarial training literature. Beyond the Lipschitz and smoothness conditions of $f$, the expansiveness parameters of $\pi$ and $J$ turn out to also influence the generalization of AT algorithms, as suggested in the generalization bound (10). This has been overlooked by the previous stability analysis as in Xing et al. (2021); Xiao et al. (2022b); Wang et al. (2024).

The behavior of the bound in (10) clearly depends on $Q(c^*)$. We now show that with additional conditions, one can choose a $c^*$ to either remove the term containing $Q(c^*)$ or make $Q(c^*)$ also vanish with $n$.

For example, if the perturbation $J$ has bounded Lipschitz constant $q^*$, that is $q_{c^*}(J) \leq q^* < \infty$ for any $c^* \geq 0$, then taking $c^* = 0$ simply results in the following bound that vanishes as $\mathcal{O}(1/n)$.

$$\mathrm{GG}_n(J, A_\pi) \leq (L_{\mathcal{X}} q^* + L_{\mathcal{W}}) \frac{2L_{\mathcal{W}}}{n\beta} \sum_{t=0}^{T} (2 + q_c(\pi)\Gamma_{\mathcal{X}}/\beta)^t \tag{11}$$

On the other hand, if the second moment of the random variable $\|A(S_*) - A(S'_*)\|$ has a fast vanishing rate with $n$, one can choose $c^*$ to decay with $n$ at a judicious choice of rate, pushing $Q(c^*)$ to vanish faster than $1/n$, resulting in the bound in the following form

$$\mathrm{GG}_n(J, A_\pi) \leq (L_{\mathcal{X}} q_{c^*}(J) + L_{\mathcal{W}}) \frac{2L_{\mathcal{W}}}{n\beta} \sum_{t=0}^{T} (2 + q_c(\pi)\Gamma_{\mathcal{X}}/\beta)^t + o(1/n) \tag{12}$$

We defer the proof of (12) to Appendix A.

**Convex loss and strongly convex loss** When $f$ is further assumed to be convex or strongly convex, a tighter UAS upper bound can be attained.

**Theorem 4.3.** *Suppose that $f(\cdot, x, y)$ is convex for any $(x, y) \in \mathcal{X} \times \mathcal{Y}$ and satisfies the conditions (4) and (5). If we run $A_\pi$ for $T$ steps with step sizes $\tau_t \leq \frac{1}{\beta}$, we have*

$$\delta_n(A_\pi) \leq \frac{2L_{\mathcal{W}}}{n\beta} \sum_{t=0}^{T} (1 + q_c(\pi)\Gamma_{\mathcal{X}}/\beta)^t$$

*If we further assume $f(\cdot, x, y)$ is $\mu-$strongly convex, we have*

$$\delta_n(A_\pi) \leq \frac{2L_{\mathcal{W}}}{n\beta} \sum_{t=0}^{T} \left(1 - \frac{\mu}{2\beta} + \Gamma_{\mathcal{X}} q_c(\pi)/\beta\right)^t$$

As shown, performing AT using convex loss functions results in a tighter upper bound compared to the non-convex functions. When $f$ is strongly convex, the bound can be tightened again. In fact, in the strongly convex case, if $q_c(\pi)$ is small enough, the UAS upper bound can be made independent of the number of iteration $T$.

**Corollary 4.4.** *Suppose that $f$ is $\mu-$strongly convex and satisfies the conditions (4) and (5). Suppose that $q_c(\pi) < \mu/(2\Gamma_{\mathcal{X}})$ and we run $A_\pi$ for $T$ steps with step sizes $\tau_t \leq \frac{1}{\beta}$, we have*

$$\delta_n(A_\pi) \leq \frac{4L_{\mathcal{W}}}{n(\mu - 2q_c(\pi)\Gamma_{\mathcal{X}})}$$

The proofs of Theorem 4.3 and Corollary 4.4 are deferred to Appendix A. Notably, when $\pi$ is chosen as the identity map, we have $q_c(\pi) = 0$ and $A_\pi$ reduces to the standard SGD algorithm. In this case, our UAS upper bounds matches the bounds in Hardt et al. (2016) up to constants.

**Comparison with existing UAS bounds for AT**  The work in Farnia & Ozdaglar (2021) derives UAS bounds for the AT-like algorithm (refer to as GDmax in their paper) under the assumption that $f$ is strongly concave in $\mathcal{X}$. Our work goes beyond this restricted setting and derive UAS bounds without this assumption. In Xing et al. (2021), the stability of AT is analyzed by treating AT as standard SGD with an adversarial loss (i.e., $f^*$) and invoke the generic bound in Bassily et al. (2020) for non-smooth losses, while assuming $f^*$ to be non-smooth. The non-smoothness is however not quantitatively characterized in their work; additionally since the bound in Bassily et al. (2020) is developed for SGD with any non-smooth convex functions, it fails to explain the notable difference between SGD and AT observed in practice. The UAS bounds proposed in Xiao et al. (2022b); Wang et al. (2024) include terms that do not vanish with increasing sample size. Our bounds overcome this limitation, vanishing with the sample size (see Appendix E for more details).

## 5 REVISIT OF PGD-BASED AT

We now discuss the AT algorithm $A_\pi$ when $\pi$ is taken as the PGD perturbation (Madry et al., 2019), which we denote by $\pi^{\mathrm{PGD}}$. To begin, associated with any $(x, y)$ and any weight parameter $w$, we define and one-step PGD map $T_{x,y,w}$ by

$$T_{x,y,w}(x') = \Pi_{\mathbb{B}_\infty(x,\epsilon)} \left[ x' + \lambda G \left( \nabla_{x'} f(w, x', y) \right) \right]$$

Here $x'$ is any point in $\mathbb{R}^d$, $G$ is a mapping from $\mathbb{R}^d$ to $\mathbb{R}^d$, possibly taking various forms, which we will specify momentarily, $\lambda$ is another step size, and $\Pi_{\mathbb{B}_\infty(x,\epsilon)} : \mathbb{R}^d \to \mathbb{B}_\infty(x, \epsilon)$ denotes the projection onto the set $\mathbb{B}_\infty(x, \epsilon)$, namely, $\Pi_{\mathbb{B}_\infty(x,\epsilon)}(x') = \arg\min_{\tilde{x} \in \mathbb{B}_\infty(x,\epsilon)} \|\tilde{x} - x'\|_2$. The $K$-step PGD perturbation $\pi^{\mathrm{PGD}}$ is then defined as the $K$-fold compositions of the (same) mapping $T_{x,y,w}$:

$$\pi^{\mathrm{PGD}}(x; y, w) := T_{x,y,w}^K(x) := \left( \underbrace{T_{x,y,w} \circ T_{x,y,w} \circ \ldots T_{x,y,w}}_{K \text{ times}} \right)(x)$$

In the well-known PGD attack (Madry et al., 2019), the mapping $G$ is taken as the sign function and is applied element-wisely on the gradients(see Wong et al. (2020); Andriushchenko & Flammarion (2020); Wang et al. (2021); Rice et al. (2020); Dong et al. (2021); Wu et al. (2020)), Theoretical analyses of PGD (as in Deng et al. (2020); Fu & Wang (2023); Bubeck et al. (2015)) often considers the "raw-gradient" version, namely taking $G$ as the identity map. In our work, we will show that the choice of $G$, this peculiar and largely overlooked building block in PGD, in fact has non-negligible impact on the generalization performance of PGD-based AT.

To begin, we assume that the gradient $\nabla_x f$ is Lipschitz, namely, that there exist positive constants $\eta$ and $\Gamma_{\mathcal{W}}$ such that for any $y \in \mathcal{Y}$, any $x, x' \in \mathcal{X}$ and any $w, w' \in \mathcal{W}$

$$\|\nabla_{x'} f(w', x', y) - \nabla_x f(w, x, y)\| \leq \eta \|x - x'\| + \Gamma_{\mathcal{W}} \|w - w'\| \tag{13}$$

**Lemma 5.1** (Expansiveness of PGD). *Suppose that $f$ satisfies the condition (13) and the mapping $G$ is $\alpha-$ Lipschitz.*

$$q_c(\pi^{\mathrm{PGD}}) \leq \min \left( \sum_{i=0}^{K-1} \mu^i \nu, \frac{2\sqrt{d}\epsilon}{c} \right)$$

*where $\nu = \lambda \alpha \Gamma_{\mathcal{W}}$ and $\mu = 1 + \lambda \alpha \eta$.*

We defer the proof to Appendix A.

For all $J$-losses for which $q_c(J)$ is uniformly bounded by $q^*$, plugging this bound to (11) immediately gives a generalization bound that vanishes as $\mathcal{O}(1/n)$. However, one of the most important $J$-loss, the one defined using sign-PGD attack, fails to satisfy this boundedness condition and the bound (11) does not apply.

To carefully study such a setting, let $J^{\mathrm{sign-PGD}} := \pi^{\mathrm{sign-PGD}}$, where $\pi^{\mathrm{sign-PGD}}$ is $\pi^{\mathrm{PGD}}$ with function $G$ taken as the sign function. We have the following results.

**Corollary 5.2.** *Let $J = J^{\mathrm{sign-PGD}}$. Suppose that for any $S$ and $S'$ with $S \simeq S'$, $\|A(S) - A(S')\| < B$ with probability 1. Under the condition of Theorem 4.1, for any $\rho > 0$, there exists some $N$ (depending on $\rho$), such that when $n > N$,*

$$\mathrm{GG}_n(J, A_\pi) < (1 - \delta_n(A_\pi)/B) L_{\mathcal{X}} \cdot 2\epsilon\sqrt{d} + \rho.$$

The proof is left in Appendix A. Note that for $J^{\text{sign}-\text{PGD}}$-loss and without additional information on $\|A(S) - A(S')\|$, it appears difficult to arrive at a generalization bound that vanishes with $n$ and the bound given here converges to a constant. Although this may not mean that AT with $J^{\text{sign}-\text{PGD}}$-loss does not have a vanishing generalization error, it nonetheless reveals certain intrinsic difficulty of generalization for this setting. Specifically, for large $n$, the perturbation radius (that defines the $J$-loss) and the input dimension appear to fight against the UAS parameter $\delta_n(A_\pi)$; when UAS parameter decreases – which pushes towards better generalization, $\epsilon\sqrt{d}$ is amplified more significantly – causing poorer generalization.

To investigate how the expansiveness property affects generalization, we consider a smooth approximation of the sign function by a $\tanh$ function, i.e., $\text{sgn}(x) \approx \tanh_\gamma(x) := \tanh(\gamma x)$. Notably, the approximation error here vanishes with increase $\gamma$. By replacing $\text{sgn}(x)$ in PGD AT with $\tanh_\gamma(x)$, we may control the expansiveness of $\pi^{\text{PGD}}$.

**Experiments**   We conduct experiment for PGD-AT when $G$ is chosen as $\tanh_\gamma$ as well as the identity map. Specially, for $\pi^{\text{PGD}}$ with different choice of $G$, we refer to it as "sign-PGD" when $G(x) = \text{sgn}(x)$, as "$\tanh_\gamma$-PGD" when $G(x) = \tanh_\gamma(x)$ and as "raw gradient (RG)-PGD" when $G(x) = x$. In all the experiments, we primarily consider the $J$-loss defined in (1) as our evaluation metric, with the loss function in $f$ taken as the 0-1 loss and refer to this metric as $J$-(0-1) loss. We mainly use $J$ from $\{\tanh_\gamma\text{-PGD}, \text{sign-PGD}, J^{\text{id}}\}$. The experiments are conducted on CIFAR-10, CIFAR-100 (Krizhevsky et al., 2009) and SVHN(Netzer et al., 2011). Our experimental setting is elaborated in Appendix B, which follows from the setting in Rice et al. (2020).

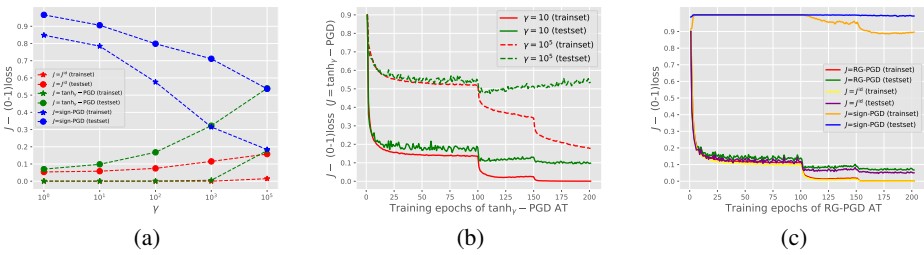

(a)                                    (b)                                    (c)

Figure 2: Experiments on CIFAR-10. (a) Models trained with $\tanh_\gamma$-PGD AT with different $\gamma$ and evaluated by $J$-(0-1) loss on the training and testing set. (b) $J$-(0-1) loss with $J = \tanh_\gamma$-PGD measured along the training trajectories of two sets of $\tanh_\gamma$-PGD AT. (c) $J$-(0-1) loss measured along the trajectory of the RG-PGD AT with different choice of $J$.

Figure 2 (a) presents the results of experiments conducted on CIFAR10, where the models are trained using $\tanh_\gamma$-PGD AT (i.e., $A_\pi$ with $\pi = \tanh_\gamma$-PGD) with various $\gamma$ values. Each model is trained for 200 epochs and is evaluated using the $J$-(0-1) loss for $J \in \{\tanh_\gamma\text{-PGD}, \text{sign-PGD}, J^{\text{id}}\}$ (distinguished by colors), where $\gamma$ matches the corresponding value in $\pi$. We use star-shaped dots and circle-shaped dots to respectively denote the $J$-(0-1) loss measured on the training set and the testing set. The gaps between each pairs of curves in the same color category then represents the generalization gap of the trained models evaluated by different $J$-(0-1) loss. By decreasing $\gamma$ in $\pi$, the generalization gaps reduce, as shown by the narrowing gaps across all pairs of the curves in the same color. The observed experimental results demonstrate that AT with less expansive $\pi$ tends to achieve a smaller generalization gap, consistent with the generalization bound of (10). Similar trends are also observed on SVHN and CIFAR100 (see Appendix D Figure 4).

Due to the mismatch between $\pi$ and $J$, the model trained by the algorithm $A_\pi$ may still have a large empirical risk $\mathbb{E}[R_S[A_\pi(S), J]]$, which in turn results in a high population risk $\mathbb{E}[R_{\mathcal{D}}[A_\pi(S), J]]$ even if the generalization gap $\text{GG}_n(J, A_\pi)$ is small. This is illustrated in Figure 2 (a) as the blue star-shaped curve consistently stays higher than the green star-shaped curve with a notably large gap. As $\gamma$ increases, the $\tanh_\gamma$ function gradually approaches the sign function, leading to an intersection of the green and the blue curves. This indicates that sign-PGD is a stronger perturbation compare to the $\tanh_\gamma$-PGD, as the model trained with $\tanh_\gamma$-PGD AT can still be vulnerable to the sign-PGD attack on the training set.

The seminal work by Tsipras et al. (2018) found that AT can negatively impact standard generalization. They constructed specific data models to demonstrate that achieving robustness and standard generalization can be inherently conflicting, suggesting an unavoidable trade-off between these two goals. This phenomenon has been extensively studied in subsequent research (Zhang et al., 2020; 2019; Yang et al., 2020; Raghunathan et al., 2020; Javanmard et al., 2020; Pang et al., 2022). Our experimental results offer further insights into this phenomenon from the perspective of algorithmic stability. Specifically, we find that the decline in standard generalization performance caused by AT can be attributed to the poor expansiveness condition of the sign-PGD method employed in AT. As shown by the trend of the red circle-shaped curve in Figure 2 (a), AT does not always harm standard generalization; a reduction in the $J^{\mathrm{id}}$-(0-1) loss is observed as $\gamma$ decreases. This suggests that the trade-off identified by Tsipras et al. (2018) might be a side effect of the sign-PGD AT and is not necessarily unavoidable.

Figure 2 (b) plots the $J$-(0-1) loss with $J = \pi$ evaluated along the trajectory of the $\tanh_\gamma$-PGD AT with $\gamma = 10$ (the solid curves) and $\gamma = 10^5$ (the dashed curves). The dashed curves exhibit a phenomenon similar to robust overfitting observed in Rice et al. (2020): after the first learning rate decay (the $100^{\mathrm{th}}$ epoch), as the training loss continuously decreases, the testing loss starts to elevate. This phenomenon does not appear in the AT with $\gamma = 10$, as shown in the trend of the solid curves. We conduct additional experiments for RG-PGD AT. As shown in Figure 2 (c), the generalization gap remains small across all groups of $J$-(0-1) loss throughout the training. Similar to the previous results, the model trained by this AT variant exhibits notable vulnerability to the sign-PGD perturbation, as indicated by the consistently high values of the orange and blue curves. These findings demonstrate that removing or altering the sign function in PGD leads to a non-negligible influence on both robust generalization and resistance to perturbations on the training set. This highlights the crucial role of the sign function in PGD-AT, which deserves a more careful and further in-depth investigation.

## 6 REVISIT OF SIGN FUNCTION IN PGD

For simplicity, we write $f(w, x, y)$ as $f(x)$ hereafter. The sign-PGD perturbation can be treated as an iterative optimization algorithm for solving the constrained optimization problem $\max_{\hat{x} \in \mathbb{B}_\infty(x, \epsilon)} f(\hat{x})$. It is related to the sign gradient methods, which has been used for different purposes, such as for training neural networks (e.g., Riedmiller & Braun (1992)) and for gradient compression (e.g., Bernstein et al. (2018)).

We now show that the sign gradient method can be viewed as a Steepest Descend (or ascend in our context) Method (SDM) w.r.t a $\infty-$norm ball (e.g., see Chapter 9.4 in Boyd & Vandenberghe (2004)). Specifically, for the loss $f(x^k)$ at the $k^{\mathrm{th}}$ iteration in SDM, it updates $x^k$ by finding a steepest ascend direction $v$ within a small neighborhood of $x^k$ such that the loss $f(x^{k+1})$ with $x^{k+1} = x^k + v$ is locally maximized. Such a neighborhood can be chosen as a $p-$norm ball around $x^k$ (i.e., $\mathbb{B}_p(x^k, \lambda_p)$) with a small radius $\lambda_p$. Finding $v$ introduces a new optimization problem: $\max_{v \in \mathbb{B}_p(x^k, \lambda_p)} f(x^k + v)$, which is then approximately solved by replacing $f(x^k + v)$ with its linear approximation around $x^k$, namely, solving $\max_{v \in \mathbb{B}_p(x^k, \lambda_p)} f(x^k) + \nabla f(x^k)^T v$ which is equivalent to solving $\max_{v \in \mathbb{B}_p(x^k, \lambda_p)} \nabla f(x^k)^T v$ whose closed form solution is

$$v^* = \lambda_p G_p(\nabla f(x^k)), \quad \text{where} \quad G_p(\nabla f(x^k)) := \frac{\mathrm{sgn}(\nabla f(x^k)) \odot |\nabla f(x^k)|^{q-1}}{\|\nabla f(x^k)\|_q^{q-1}} \quad (14)$$

where we require $1/p + 1/q = 1$. The operator $\odot$ denotes the element-wise product. The closed form (14) then gives the following updating rule of SDM as

$$x^{k+1} = x^k + \lambda G_p(\nabla f(x^k))$$

As a special case, when $p = 1$ with $q = \infty$, SMD turns into the coordinate gradient method with $G_1(\nabla f(x^k)) = \mathrm{sgn}(\max_i \nabla f(x^k)[i])e_i$ and $i = \arg\max_j |\nabla f(x^k)[j]|$, where $e_i$ denotes the standard basis vector. When $p = q = 2$, we have $G_2(\nabla f(x^k)) = \nabla f(x^k)/\|\nabla f(x^k)\|_2$

When $p = \infty$ with $q = 1$, the mapping $G_\infty$ reduces to the sign function, indicating that the sign-PGD is indeed a (projected) SDM w.r.t $\mathbb{B}_\infty(x^k, \lambda_\infty)$. It is then curious to investigate *the generalization performance of the model trained by AT using the $G_p-$PGD with $p \neq \infty$.*[1] We conduct

---

[1]Note that in the $G_p-$PGD we still consider projecting onto $\mathbb{B}_\infty(x, \epsilon)$ when $p$ is taken other than $\infty$.

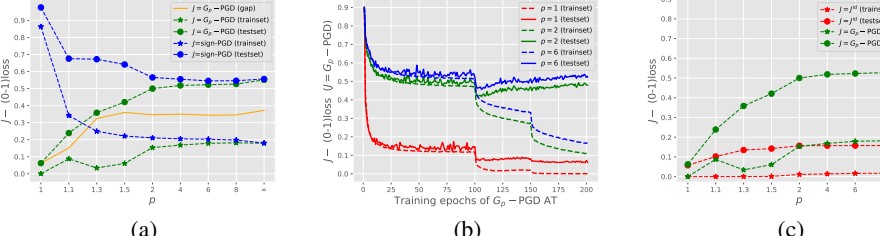

Figure 3: Experiments for $G_p$-PGD AT: (a) Model trained with various $p$ values and evaluated by $J$-(0-1) loss with $J = \pi$ and $J = $ sign-PGD. (b) Training curves of the AT with various $p$ values. (c) Standard generalization performance of the models trained by the AT, where the green curves are copied from (a) for a clearer presentation.

experiments for the $G_p$-PGD-based AT following the same experimental setting as in the previous section, except that $\lambda_p$ is adjusted to maintain the same volume of the balls $\mathbb{B}_p(0, \lambda_p)$ across different $p$ values (details in Appendix C). Figure 3 (a) presents the experimental results on CIFAR10 (results on the other datasets are in Appendix D Figure 5 and 6). The models are trained by $A_\pi$ with $\pi = G_p$-PGD for various $p$ and are evaluated by the $J$-(0-1) loss with $J = \pi$ (green curves) as well as $J =$ sign-PGD (blue curves). The yellow curve represents the generalization gap for models trained with $G_p$-PGD. As shown, a larger $p$ tends to result in larger generalization gaps. Indeed, nearly all $G_p$-PGD with $p \geq 1.3$ cause notably overfitting in AT with generalization gaps exceeding 30%. The consistently higher position of blue star-shaped curves over the green star-shaped curve also suggests that sign-PGD is the strongest perturbations among the $G_p$-PGD. Figure 3 (b) further exhibits the overfitting in $G_p$-PGD AT by plotting training curves for $p = \{1, 2, 6\}$, where continued training causes a rise of the testing errors (the blue and green curves), in contrast with the red curves, which demonstrate a good generalization. Figure 3 (c) shows how the $G_p$-PGD AT affect standard generalization where the red curves deontes the $J$-(0-1) loss with $J = J^{\text{id}}$ and the green curves are copied from Figure 3 (a) for a clearer comparison. An enlarging standard generalization gap is also observed in $G_p$-PGD AT with larger $p$.

The observed overfitting caused by the $G_p$−PGD family is potentially attributed to that nearly all the members in $\{G_p : p \in [1, \infty]\}$ have a poor Lipschitzness, as shown in the following lemma, which leads to a bad expansiveness of $G_p$-PGD.

**Lemma 6.1.** *Consider the mapping $G_p : \mathbb{R}^d \to \mathbb{R}^d$ specified in (14) with $p \in [1, \infty]$. Let $\mathcal{I} := \{1, \cdots, d\}$. If $G_p$ is $\alpha_p-$Lipschitz over the set $H(r) \subseteq \mathbb{R}^d$ with $H(r) := \{x \in \mathbb{R}^d : \min_{i \in \mathcal{I}} |x[i]| \geq r\}$ for some $r > 0$, then we have*

$$\alpha_p \geq \frac{1}{r d^{\frac{1}{p}}}$$

We defer the proof in Appendix A. This lower bound also implies that $\alpha_p$ is unbounded in $\mathbb{R}^d$, noting that the lower bound approaches infinity as $r \to 0$. Except for this extreme case, it is reasonable to assume that the gradients $\nabla_x f(x)$ lies in a set $H(r)$ with sufficiently small $r$ where all the members in $\{G_p : p \in [1, \infty]\}$ have a bounded but large Lipschitz constant. Noteworthy, the lower bound increases, as $p$ ranges from 1 to infinity, suggesting that the increased generalization gap in Figure 3 (a) is attributed to the increasing expansiveness of $G_p$−PGD caused by the rise in $\alpha_p$.

## 7 LIMITATIONS

The main limitation of this work is that we have only developed an upper bound for the generalization of AT algorithms. Like all theoretical results based on upper bounds, they are adequate for understanding performance guarantees but may be inadequate to explain poor generalization. Nonetheless, our experimental results have suggested that our upper bound may well explain robust overfitting.

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

## A    PROOFS

**Proof of Lemma 3.2**

$$\Delta_n(A, f_J) = \sup_{S \simeq S'} \sup_{(x,y) \in \mathcal{X} \times \mathcal{Y}} \mathbb{E}_A[f(A(S), J(x; y, A(S)), y) - f(A(S'); J(x; y, A(S')), y)] \tag{15}$$

$$= \sup_{(x,y) \in \mathcal{X} \times \mathcal{Y}} \mathbb{E}_A[f(A(S_*), J(x; y, A(S_*)), y) - f(A(S'_*); J(x; y, A(S'_*)), y)] \tag{16}$$

$$\leq \sup_{(x,y) \in \mathcal{X} \times \mathcal{Y}} \mathbb{E}_A[L_{\mathcal{X}} \| J(x; y, A(S_*)) - J(x; y, A(S'_*)) \| + L_{\mathcal{W}} \| A(S_*) - A(S'_*) \|] \tag{17}$$

$$= \sup_{(x,y) \in \mathcal{X} \times \mathcal{Y}} \mathbb{E}_A L_{\mathcal{X}} \| J(x; y, A(S_*)) - J(x; y, A(S'_*)) \| + L_{\mathcal{W}} \mathbb{E}_A \| A(S_*) - A(S'_*) \| \tag{18}$$

$$\leq \sup_{(x,y) \in \mathcal{X} \times \mathcal{Y}} \mathbb{E}_A L_{\mathcal{X}} \| J(x; y, A(S_*)) - J(x; y, A(S'_*)) \| + L_{\mathcal{W}} \sup_{S \simeq S'} \mathbb{E}_A \| A(S) - A(S') \| \tag{19}$$

The inequality (17) is derived based on the condition (4). We now deal with the first term in (19).

For shorter notation, let $D(S_*, S'_*) := \| A(S_*) - A(S'_*) \|$. For any number $c^* \geq 0$, let $Q(S_*, S'_*; c^*) := \Pr(D(S_*, S'_*) < c^*)$. For any $x, y$ we have

$$\mathbb{E}_A[L_{\mathcal{X}} \| J(x; y, A(S_*)) - J(x; y, A(S'_*)) \|] \tag{20}$$

$$= (1 - Q(S_*, S'_*; c^*)) \mathbb{E}_A [L_{\mathcal{X}} \| J(x; y, A(S_*)) - J(x; y, A(S'_*)) \| \mid D(S_*, S'_*) \geq c^*] \tag{21}$$

$$+ Q(S_*, S'_*; c^*) \mathbb{E}_A[L_{\mathcal{X}} \| J(x; y, A(S_*)) - J(x; y, A(S'_*)) \| \mid D(S_*, S'_*) < c^*] \tag{22}$$

$$\leq (1 - Q(S_*, S'_*; c^*)) \mathbb{E}_A[q_{c^*}(J) L_{\mathcal{X}} D(S_*, S'_*) \mid D(S_*, S'_*) \geq c^*] + Q(S_*, S'_*; c^*) L_{\mathcal{X}} 2\epsilon \sqrt{d} \tag{23}$$

$$\leq q_{c^*}(J) L_{\mathcal{X}} \mathbb{E}_A D(S_*, S'_*) + Q(S_*, S'_*; c^*) L_{\mathcal{X}} 2\epsilon \sqrt{d} \tag{24}$$

$$\leq q_{c^*}(J) L_{\mathcal{X}} \sup_{S \simeq S'} \mathbb{E}_A D(S, S') + Q(S_*, S'_*; c^*) L_{\mathcal{X}} 2\epsilon \sqrt{d} \tag{25}$$

The derivation above start by splitting the expectation into two conditional expectations conditioned on two complementary events (see the terms (21) and (22)) and then utilize the $c-$expansiveness property of $J$ as well as the condition that $J(x, y, w) \in \mathbb{B}_\infty(x, \epsilon)$ to individually derive the first and second terms in (23). Plug the final expression above back in (19), the lemma is proved.    $\square$

**Proof of the Theorem 4.1**    Consider the AT algorithm specified in (7) and (8). For two datasets $S$ and $S'$ differing in only one sample and respectively containing $n$ samples, let $\{w_t\}_{t=1}^T$ and $\{w'_t\}_{t=1}^T$ respectively denote the sequences of model parameters generated by running AT on $S$ and $S'$ for $T$ iterations. Let $c$ denote the smallest non-zero value of $\| w_t - w'_t \|$ across $t$ and across the randomness of $A$ when running AT algorithm $A$ on $S$ and $S'$. (Note that such a choice of $c$ may be overly pessimistic, but it suffices to obtain the desired rate of vanishing of the generalization bound in this theorem). For arbitrary iteration $t \in \{1, \cdots, T - 1\}$, we have

$$\mathbb{E}\| w_{t+1} - w'_{t+1} \|$$
$$\leq \mathbb{E}\| w_t - \tau_t \nabla_{w_t} f(w_t, \pi(x; y, w_t), y) + \tau_t \nabla_{w'_t} f(w'_t, \pi(x; y, w_t), y) - w'_t \|$$
$$+ \mathbb{E}\| \tau_t \nabla_{w'_t} f(w'_t, \pi(x'; y', w'_t), y') - \tau_t \nabla_{w'_t} f(w'_t, \pi(x; y, w_t), y) \| \tag{26}$$

Here the expectation is taken over all the randomness in $w_t$ and $w'_t$. We use $(x, y)$ and $(x', y')$ respectively to denote the samples selected by the AT algorithm from $S$ and $S'$ at the iteration $t$. Inequality (26) is derived by adding and subtracting the term $\tau_t \nabla_{w'_t} f(w'_t, \pi(x; y, w_t), y)$ and then applying the triangle inequality. For the first term in (26), we have that

$$\mathbb{E}\| w_t - \tau_t \nabla_{w_t} f(w_t, \pi(x; y, w_t), y) + \tau_t \nabla_{w'_t} f(w'_t, \pi(x; y, w_t), y) - w'_t \|$$
$$\leq \mathbb{E}\| w_t - w'_t \| + \tau_t \beta \mathbb{E}\| w_t - w'_t \| \tag{27}$$

by utilizing the triangle inequality and the condition (5). To deal with the second term in (26), we consider that at each iteration, with probability $1 - 1/n$ the samples selected by AT respectively

from $S$ and $S'$ are the same. We have

$$\mathbb{E}\|\tau_t \nabla_{w'_t} f\left(w'_t, \pi(x'; y', w'_t), y'\right) - \tau_t \nabla_{w'_t} f\left(w'_t, \pi(x; y, w_t), y\right)\|$$

$$\leq \left(1 - \frac{1}{n}\right) \tau_t \Gamma_{\mathcal{X}} \mathbb{E}\|\pi(x; y, w'_t) - \pi(x; y, w_t)\| + \frac{2\tau_t L_{\mathcal{W}}}{n} \tag{28}$$

$$\leq \left(1 - \frac{1}{n}\right) \tau_t \Gamma_{\mathcal{X}} q_c(\pi) \mathbb{E}\|w_t - w'_t\| + \frac{2\tau_t L_{\mathcal{W}}}{n} \tag{29}$$

The first term in (28) and (30) make use of the condition (5) and then the expansiveness condition of $\pi$. Since $f$ is $L_{\mathcal{W}}-$ Lipschitz w.r.t $\mathcal{W}$, we have $\|\nabla_w f(w; x, y)\| \leq L_{\mathcal{W}}$ for $\forall x, y, w$. The second term in (28) then follows.

Putting together and considering the step sizes $\tau_t \leq \frac{1}{\beta}$, we have

$$\mathbb{E}\|w_{t+1} - w'_{t+1}\|$$

$$\leq \left(1 + \beta\tau_t + (1 - 1/n)\Gamma_{\mathcal{X}} q_c(\pi)\tau_t\right) \mathbb{E}\|w_t - w'_t\| + \frac{2\tau_t L_{\mathcal{W}}}{n} \tag{30}$$

$$\leq \left(1 + \beta\tau_t + \Gamma_{\mathcal{X}} q_c(\pi)\tau_t\right) \mathbb{E}\|w_t - w'_t\| + \frac{2\tau_t L_{\mathcal{W}}}{n} \tag{31}$$

$$\leq \left(2 + \Gamma_{\mathcal{X}} q_c(\pi)/\beta\right) \mathbb{E}\|w_t - w'_t\| + \frac{2L_{\mathcal{W}}}{n\beta} \tag{32}$$

Unravelling the recursion, we have

$$\mathbb{E}\|w_T - w'_T\| \leq \frac{2L_{\mathcal{W}}}{n\beta} \sum_{t=0}^{T} \zeta^t \tag{33}$$

where we take $\zeta = 2 + \Gamma_{\mathcal{X}} q_c(\pi)/\beta$. $\qquad\square$

**Proof of (12)** Let $a > 2$ be a constant. For shorter notation let $Z = \|A_\pi(S_*) - A_\pi(S'_*)\|$. We will show that if the second moment $\mathbb{E}Z^2 = \mathcal{O}(\frac{1}{n^a})$, we can take $c^* = \mathbb{E}Z - t$ with $t = \Omega(\frac{1}{n^b})$ and $b \in (1, a/2)$, such that the probability $Q(c^*)$ decay at the rate of $\frac{1}{n^{a-2b}}$. This is due to that

$$Q(c^*) = \Pr\left[Z \leq c^*\right] \tag{34}$$

$$= \Pr\left[Z \leq \mathbb{E}Z - t\right] \tag{35}$$

$$\leq \Pr\left[t \leq |Z - \mathbb{E}Z|\right] \tag{36}$$

$$\leq \frac{\mathrm{Var}(Z)}{t^2} \tag{37}$$

$$\leq \frac{\mathbb{E}Z^2}{t^2} \tag{38}$$

$$\leq \mathcal{O}\left(\frac{1/n^a}{1/n^{2b}}\right) = \mathcal{O}\left(\frac{1}{n^{a-2b}}\right) \tag{39}$$

where the inequality (37) is based on the Chebyshev's inequality. Note that such a choice of $t$ will guarantee that $c^* > 0$ such that the derivation above is nontrivial. This is because Theorem 4.1 implies that $\mathbb{E}Z \leq \delta_n(A_\pi) = \mathcal{O}(\frac{1}{n})$ and therefore $c^* = \mathcal{O}(\frac{1}{n} - \frac{1}{n^b})$. Taking $b > 1$ guarantees that $c^* > 0$.

**Proof of the Theorem 4.3 and Corollary 4.4** The proof is based on a slight modification of the proof in Theorem 4.1. We start from the inequality (26). For the first term in (26), since that the loss function $f$ is convex and $\tau_t \leq 1/\beta < 2/\beta$, according to Lemma 3.7.2 in Hardt et al. (2016), we have

$$\mathbb{E}\|w_t - \tau_t \nabla_{w_t} f\left(w_t, \pi(x; y, w_t), y\right) + \tau_t \nabla_{w'_t} f\left(w'_t, \pi(x; y, w_t), y\right) - w'_t\|$$

$$\leq \mathbb{E}\|w_t - w'_t\| \tag{40}$$

When $f$ is further assumed to be $\mu-$ strongly convex, we have that $\mu \leq \beta$ since $f$ is also $\beta-$smooth, implying that $\tau_t \leq \frac{1}{\beta} \leq \frac{2}{\beta+\mu}$. According to Lemma 3.7.3 in Hardt et al. (2016), we have inequality

(41) as

$$\mathbb{E}\|w_t - \tau_t \nabla_{w_t} f(w_t, \pi(x; y, w_t), y) + \tau_t \nabla_{w'_t} f(w'_t, \pi(x; y, w_t), y) - w'_t\|$$

$$\leq \left(1 - \frac{\beta\mu\tau_t}{\beta + \mu}\right) \mathbb{E}\|w_t - w'_t\| \tag{41}$$

$$\leq \left(1 - \frac{1}{2}\tau_t\mu\right) \mathbb{E}\|w_t - w'_t\| \tag{42}$$

In fact, since $\mu \leq \beta$, we also have $1 \leq \frac{2\beta}{\beta+\mu}$ and thus $\tau_t\mu \leq \frac{2\tau_t\mu\beta}{\beta+\mu}$ with $\tau_t\mu \leq 1$. The inequality (41) can be further simplified as (42).

The second term in (26) follows the same derivation as in the proof of Theorem 4.1. Putting together, when $f$ is convex, we have

$$\mathbb{E}\|w_{t+1} - w'_{t+1}\|$$

$$\leq (1 + \Gamma_{\mathcal{X}} q_c(\pi)\tau_t) \mathbb{E}\|w_t - w'_t\| + \frac{2\tau_t L_{\mathcal{W}}}{n} \tag{43}$$

$$\leq (1 + \Gamma_{\mathcal{X}} q_c(\pi)/\beta) \mathbb{E}\|w_t - w'_t\| + \frac{2L_{\mathcal{W}}}{n\beta} \tag{44}$$

when $f$ is $\mu-$ strongly convex, we have

$$\mathbb{E}\|w_{t+1} - w'_{t+1}\|$$

$$\leq \left(1 - \frac{1}{2}\tau_t\mu + \Gamma_{\mathcal{X}} q_c(\pi)\tau_t\right) \mathbb{E}\|w_t - w'_t\| + \frac{2\tau_t L_{\mathcal{W}}}{n} \tag{45}$$

$$\leq \left(1 - \frac{\mu}{2\beta} + \Gamma_{\mathcal{X}} q_c(\pi)/\beta\right) \mathbb{E}\|w_t - w'_t\| + \frac{2L_{\mathcal{W}}}{n\beta} \tag{46}$$

Unravelling the recursion, we have

$$\mathbb{E}\|w_T - w'_T\| \leq \frac{2L_{\mathcal{W}}}{n\beta} \sum_{t=0}^{T} \zeta^t \tag{47}$$

with $\zeta = 1 + \Gamma_{\mathcal{X}} q_c(\pi)/\beta$ when $f$ is convex and $\zeta = 1 - \frac{\mu}{2\beta} + \Gamma_{\mathcal{X}} q_c(\pi)/\beta$ when $f$ is $\mu-$ strongly convex. For the strongly convex case, if we let $q_c(\pi) < \frac{\mu}{2\Gamma_{\mathcal{X}}}$, we have $\zeta < 1$. In this case, the geometric series $\sum_{t=0}^{T} \zeta^t$ converges as $T \to \infty$ and entails a closed form. The bound in (47) can therefore be further simplified as

$$\mathbb{E}\|w_T - w'_T\| \leq \frac{2L_{\mathcal{W}}}{n\beta} \sum_{t=0}^{T} \zeta^t$$

$$\leq \frac{2L_{\mathcal{W}}}{n\beta} \sum_{t=0}^{\infty} \zeta^t \tag{48}$$

$$= \frac{2L_{\mathcal{W}}}{n\beta} \frac{1}{1-\zeta} \tag{49}$$

$$= \frac{4L_{\mathcal{W}}}{n(\mu - 2q_c(\pi)\Gamma_{\mathcal{X}})} \tag{50}$$

This derives the bound in Corollary 4.4. $\qquad\square$

**Proof of Lemma 5.1** To establish the proof, we first discuss the expansive property of the one step PGD perturbation $T$. For arbitrary $\hat{x} \in \mathcal{X}$, we have

$$\|T_{x,y}(\hat{x}; w) - T_{x,y}(\hat{x}; w')\| \tag{51}$$

$$= \left\|\Pi_{\mathbb{B}_\infty(x,\epsilon)} [\hat{x} + \lambda G(\nabla_{\hat{x}} f(w, \hat{x}, y))] - \Pi_{\mathbb{B}_\infty(x,\epsilon)} [\hat{x} + \lambda G(\nabla_{\hat{x}} f(w', \hat{x}, y))]\right\| \tag{52}$$

$$\leq \lambda \|G(\nabla_{\hat{x}} f(w, \hat{x}, y)) - G(\nabla_{\hat{x}} f(w', \hat{x}, y))\| \tag{53}$$

$$\leq \lambda\alpha \|\nabla_{\hat{x}} f(w, \hat{x}, y) - \nabla_{\hat{x}} f(w', \hat{x}, y)\| \tag{54}$$

$$\leq \lambda\alpha\Gamma_{\mathcal{W}} \|w - w'\| \tag{55}$$

The inequality (53) is due to that the projection operation $\Pi_{\mathbb{B}_\infty(x,\epsilon)}$ is 1-expansive. The inequalities (54) and (55) are derived based on the Lipschitz condition of $G$ and $\nabla_x f$.

For fixed $w \in \mathcal{W}$, we have for arbitrary $x', x'' \in \mathcal{X}$

$$\|T_{x,y}(x'; w) - T_{x,y}(x''; w)\| \tag{56}$$
$$= \left\| \Pi_{\mathbb{B}_\infty(x,\epsilon)} \left[ x' + \lambda G \left( \nabla_{x'} f(w, x', y) \right) \right] - \Pi_{\mathbb{B}_\infty(x,\epsilon)} \left[ x'' + \lambda G \left( \nabla_{x''} f(w, x'', y) \right) \right] \right\| \tag{57}$$
$$\leq \left\| x' + \lambda G \left( \nabla_{x'} f(w, x', y) \right) - x'' + \lambda G \left( \nabla_{x''} f(w, x'', y) \right) \right\| \tag{58}$$
$$\leq \| x' - x'' \| + \lambda \alpha \left\| \nabla_{x'} f(w, x', y) - \nabla_{x''} f(w, x'', y) \right\| \tag{59}$$
$$\leq (1 + \lambda \alpha \eta) \| x' - x'' \| \tag{60}$$

The derivation here follows the similar idea as above, utilizing the 1-expansiveness condition of $\Pi_{\mathbb{B}_\infty(x,\epsilon)}$ as well as the Lipschitz condition of $G$ and the smoothness condition of $f$ w.r.t $\mathcal{X}$.

We now derive the upper bound for the expansiveness of $\pi^{\mathrm{PGD}}$. With a little abuse of notation, let $x_K = T_{x,y}^K(x; w)$ and similarly $x_K' = T_{x,y}^K(x; w')$. For shorter notation, let $\nu = \lambda \alpha \Gamma_{\mathcal{W}}$ and $\mu = 1 + \lambda \alpha \eta$.

$$\|\pi^{\mathrm{PGD}}(x; y, w) - \pi^{\mathrm{PGD}}(x; y, w')\| \tag{61}$$
$$= \|T_{x,y}^K(x; w) - T_{x,y}^K(x; w')\| \tag{62}$$
$$= \|T_{x,y}(x_{K-1}; w) - T_{x,y}(x_{K-1}'; w')\| \tag{63}$$
$$\leq \|T_{x,y}(x_{K-1}; w) - T_{x,y}(x_{K-1}; w')\| + \|T_{x,y}(x_{K-1}; w') - T_{x,y}(x_{K-1}'; w')\| \tag{64}$$
$$\leq \pi \|w - w'\| + \mu \|x_{K-1} - x_{K-1}'\| \tag{65}$$
$$= \pi \|w - w'\| + \mu \|T_{x,y}(x_{K-2}; w) - T_{x,y}(x_{K-2}'; w')\| \tag{66}$$
$$\leq \sum_{i=0}^{K-1} \mu^i \nu \|w - w'\| \tag{67}$$

Note that the bound (67) holds for any choice of $w, w'$. On the other hand, using the condition that $T_{x,y}(\hat{x}; w) \in \mathbb{B}_\infty(x, \epsilon)$, we can derive that for any $w, w' \in \mathcal{W}$ with $\|w - w'\| > c$,

$$\|T_{x,y}(\hat{x}; w) - T_{x,y}(\hat{x}; w')\| \leq 2\sqrt{d}\epsilon = \frac{2\sqrt{d}\epsilon}{\|w - w'\|} \|w - w'\| \leq \frac{2\sqrt{d}\epsilon}{c} \|w - w'\| \tag{68}$$

Putting together, we have

$$q_c(\pi^{\mathrm{PGD}}) \leq \min \left( \sum_{i=0}^{K-1} \mu^i \nu, \frac{2\sqrt{d}\epsilon}{c} \right) \tag{69}$$

This completes the proof. $\qquad \square$

**Proof of Corollary 5.2**  We first establish the following result.

For any non-negative random variable $Z$ bounded below $B$ and any $c^* > 0$,

$$\Pr[Z \leq c^*] \leq \frac{B - \mathbb{E}(Z)}{B - c^*} \tag{70}$$

This result simply follows from $\Pr[Z \leq c^*] = \Pr[B - Z \geq B - c^*]$ and applying the Markov Inequality to random variable $B - Z$.

Now let $Z = A(S) - A(S')$ and $c^* = Bn^{-1/2}$ in Theorem 4.2. The second term in bound of Theorem 4.2 then reduces to $\left( 1 - \frac{\sup_{S \simeq S'} \mathbb{E}\|A(S) - A(S')\|}{B(1 - n^{-1/2})} \right) L_{\mathcal{X}} \cdot 2\epsilon\sqrt{d}$, which converges to $\left( 1 - \frac{\sup_{S \simeq S'} \mathbb{E}\|A(S) - A(S')\|}{B} \right) L_{\mathcal{X}} \cdot 2\epsilon\sqrt{d}$ with $n$. It can be verified that the first term in the bound of Theorem 4.2 vanishes with $n$ (as $n^{-1/2}$). The corollary then follows. $\qquad \square$.

**Proof of Lemma 6.1** The proof is established by noticing that all members in the set $\tilde{H}(r) := \{x \in \mathbb{R}^d : |x[i]| = r, \forall i \in \mathcal{I}\}$ achieves $1/(rd^{\frac{1}{p}})$−Lipschitz and thus the Lipschitz constant over $H(r)$ is greater than it. Specifically, for any $x, \hat{x} \in \tilde{H}(r)$ with $x \neq \hat{x}$, let $\mathcal{I}_- := \{i \in \mathcal{I} : \mathrm{sgn}(x[i]) \neq \mathrm{sgn}(\hat{x}[i])\}$ and $\mathcal{I}_+ := \mathcal{I} - \mathcal{I}_-$. We have

$$\|G(x) - G(\hat{x})\|_2 \tag{71}$$

$$= \left\| \frac{\mathrm{sgn}(x) \odot |x|^{q-1}}{\|x\|_q^{q-1}} - \frac{\mathrm{sgn}(\hat{x}) \odot |\hat{x}|^{q-1}}{\|\hat{x}\|_q^{q-1}} \right\|_2 \tag{72}$$

$$= \left( \sum_{i=1}^d \left| \frac{\mathrm{sgn}(x[i])|x[i]|^{q-1}}{\|x\|_q^{q-1}} - \frac{\mathrm{sgn}(\hat{x}[i])|\hat{x}[i]|^{q-1}}{\|\hat{x}\|_q^{q-1}} \right|^2 \right)^{\frac{1}{2}} \tag{73}$$

$$= \left( \sum_{j \in \mathcal{I}_+} \left| \frac{\mathrm{sgn}(x[j])|x[j]|^{q-1}}{\|x\|_q^{q-1}} - \frac{\mathrm{sgn}(\hat{x}[j])|\hat{x}[j]|^{q-1}}{\|\hat{x}\|_q^{q-1}} \right|^2 + \sum_{k \in \mathcal{I}_-} \left| \frac{\mathrm{sgn}(x[k])|x[k]|^{q-1}}{\|x\|_q^{q-1}} - \frac{\mathrm{sgn}(\hat{x}[k])|\hat{x}[k]|^{q-1}}{\|\hat{x}\|_q^{q-1}} \right|^2 \right)^{\frac{1}{2}} \tag{74}$$

$$= \left( \sum_{k \in \mathcal{I}_-} \left| \frac{2r^{q-1}}{r^{q-1}d^{\frac{1}{p}}} \right|^2 \right)^{\frac{1}{2}} \tag{75}$$

$$= \sqrt{|\mathcal{I}_-|} \frac{2}{d^{\frac{1}{p}}} \tag{76}$$

where $|\mathcal{I}_-|$ denotes the cardinality of the set $\mathcal{I}_-$. The equality (75) is derived by noting that the first term in (74) is zero since $|x[j]| = |\hat{x}[j]|$ and $\mathrm{sgn}(|x[j]|) = \mathrm{sgn}(|\hat{x}[j]|)$ for each $j \in \mathcal{I}_+$ and noting that $\|x\|_q = rd^{\frac{1}{q}}$ for any $x \in \tilde{H}(r)$. The power term $\frac{q-1}{q}$ is replaced by $\frac{1}{p}$ since $1/q + 1/p = 1$. We also have

$$\|x - \hat{x}\|_2 \tag{77}$$

$$= \left( \sum_{i=1}^d |x[i] - \hat{x}[i]|^2 \right)^{\frac{1}{2}} \tag{78}$$

$$= \left( \sum_{j \in \mathcal{I}_+} |x[j] - \hat{x}[j]|^2 + \sum_{k \in \mathcal{I}_-} |x[k] - \hat{x}[k]|^2 \right)^{\frac{1}{2}} \tag{79}$$

$$= \left( \sum_{k \in \mathcal{I}_-} |2r|^2 \right)^{\frac{1}{2}} \tag{80}$$

$$= 2r\sqrt{|\mathcal{I}_-|} \tag{81}$$

Putting together, we have that for any $x, \hat{x} \in \tilde{H}(r)$ with $x \neq \hat{x}$,

$$\frac{\|G(x) - G(\hat{x})\|_2}{\|x - \hat{x}\|_2} = \frac{1}{rd^{\frac{1}{p}}} \leq \sup_{\substack{x', x'' \in Q(r) \\ x' \neq x''}} \frac{\|G(x') - G(x'')\|_2}{\|x' - x''\|_2} = \alpha_p \tag{82}$$

This completes the proof. $\qquad\square$

## B HYPER-PARAMETER SETTINGS FOR THE EXPERIMENTS

In our experiments, we follow the settings in Rice et al. (2020): The perturbation radius is set to be $\epsilon = 8/255$ w.r.t the $\infty$−norm for the three datasets. The pre-activation ResNet 18 (PRN-18) model (He et al., 2016) is used for CIFAR-10 and SVHN. The Wide ResNet 34 (WRN-34) model (Zagoruyko & Komodakis, 2016) is used for CIFAR-100. We set $K = 10$ for all the PGD variants with $\lambda = 2/255$ on CIFAR-10 and CIFAR-100, and set $\lambda = 1/255$ for SVHN. The initial learning

rate of AT is set to be 0.1 for CIFAR-10 and CIFAR-100 and set to be 0.01 for SVHN. The learning rate is decayed by 0.1 at the 100[th] and the 150[th] epoch of the training. The batch size is set to be 128 and a weight decay of $5 \times 10^{-4}$ is used for all the experiments. The experiments are conducted on our internal GPU clusters. Training PRN-18 on CIFAR-10 and SVHN for 200 epochs spends around 18 hours with two NVIDIA V100 GPUs, and training WRN-34 on CIFAR-100 requires around three days to complete with the same computing resources.

## C  COMPUTING $\lambda_p$

The volume of $\mathbb{B}_p(0, \lambda_p)$ is computed by

$$\text{vol}\left(\mathbb{B}_p(0, \lambda_p)\right) = \frac{\left(2\Gamma\left(\frac{1}{p} + 1\right)\right)^d}{\Gamma\left(\frac{d}{p} + 1\right)} \lambda_p^d \tag{83}$$

Here $\Gamma(\cdot)$ denotes the Euler's gamma function. For $p$ other than $\infty$, to make $\text{vol}\left(\mathbb{B}_p(0, \lambda_p)\right) = \text{vol}\left(\mathbb{B}_\infty(0, \lambda_\infty)\right)$, we have

$$\lambda_p = \exp\left\{\frac{1}{d}\ln\Gamma(\frac{d}{p} + 1) + \ln\frac{\lambda_\infty}{\Gamma(\frac{1}{p} + 1)}\right\} \tag{84}$$

In the experiments, the value of $\lambda_\infty$ (i.e., the step size for the sign-PGD) is set to be the same as in Section 5 and values for other $\lambda_p$ is computed from (84).

# D OMITTED FIGURES

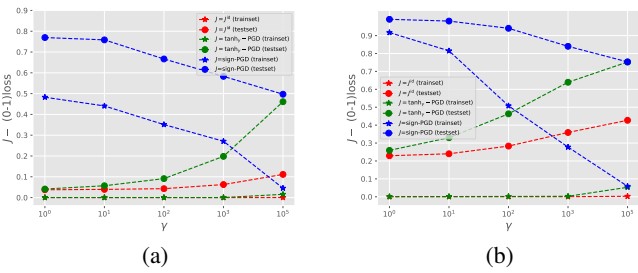

(a)             (b)

Figure 4: Experiments in Figure 2 reproduced on SVHN and CIFAR-100.

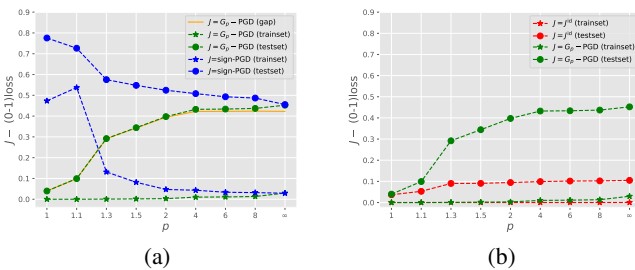

(a)             (b)

Figure 5: Experiments in Figure 3 reproduced on SVHN.

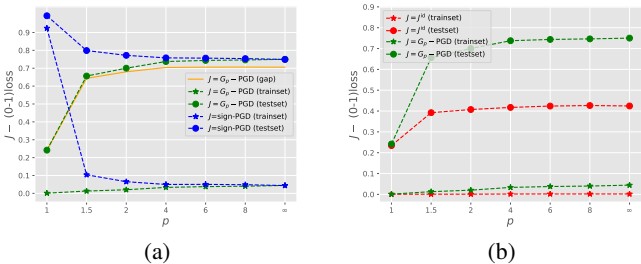

(a)             (b)

Figure 6: Experiments in Figure 3 reproduced on CIFAR-100.

# E COMPARISON WITH THE WORKS IN XIAO ET AL. (2022B) AND WANG ET AL. (2024)

Since the work of Wang et al. (2024) is built upon the framework in Xiao et al. (2022b), we here only presents the connections and differences between Xiao et al. (2022b) and our work.

**Summary of generalization bounds in Xiao et al. (2022b)** First we would like to note that the problem setting in this paper includes the setting in Xiao et al. (2022b) as a special case. Specifically, the generalization gap discussed in Xiao et al. (2022b) corresponds to the generalization gap $\mathrm{GG}_n(J^*, A_{J^*})$ defined in this paper, where the perturbations in both $J-$loss and the AT algorithm are taken as the optimal adversarial perturbation $J^*$.

This work and Xiao et al. (2022b) both take the Lipschitzness and smoothness conditions of the standard loss $f$ as the starting point, but derive generalization bounds from different perspectives:

the work in Xiao et al. (2022b) defines and proposes to study the $\eta-$approximate smoothness of the adversarial loss ( $f^*$ in our notation) and derive generalization bounds based on this quantity. This work defines the notion of $c-$expansiveness of the perturbation operator (e.g., $J^*$) and show how this quantity affects generalization performance of AT.

For completeness, we here present the definition of $\eta-$approximate smoothness, rewrite the Definition 4.1 of Xiao et al. (2022b) using the notations in this paper.

**Definition E.1** ($\eta-$approximate smoothness Xiao et al. (2022b))**.** A loss function $f_J$ is called $\eta-$approximately $\beta-$gradient Lipschitz if there exists $\beta > 0$ and $\eta > 0$ such that for any $(x, y) \in \mathcal{X} \times \mathcal{Y}$ and for any $w_1, w_2 \in \mathcal{W}$ we have

$$\|\nabla f_J(w_1, x, y) - \nabla f_J(w_2, x, y)\| \leq \beta \|w_1 - w_2\| + \eta \tag{85}$$

The work in Xiao et al. (2022b) then derives generalization bounds for loss functions that are $\eta-$approximately smooth. For example, after replacing the notations in Xiao et al. (2022b) with ours, Theorem 5.1 of Xiao et al. (2022b) shows that if $f_J$ is $\eta-$approximately $\beta-$gradient Lipschitz, convex in $w$ for all $(x, y)$ and the standard loss $f$ satifies the same Lipschitz condition in (6) of this paper (or Assumption 4.1. in Xiao et al. (2022b)), then their bound in Theorem 5.1 becomes

$$\mathrm{GG}_n(J, A_J) \leq \frac{L_{\mathcal{W}}}{\beta} \eta T + \frac{2L_{\mathcal{W}}^2}{n\beta} T$$

The authors of Xiao et al. (2022b) show that the adversarial loss $f^*$ satisfies $\eta$-approximately $\beta$-gradient Lipschitz with $\eta = 2\Gamma_{\mathcal{X}}\epsilon$ so that the generalization bound above gives their generalization bound for adversarial training. In their determination of the $\eta$ parameter, they have assumed that the standard loss $f$ satisfies certain Lipschitz and smoothness condition; this condition is effectively equivalent to the condition (5) in this paper.

It is worth noting that the generalization bounds derived based on the approximate smoothness parameter $\eta$ contain a term unrelated to the sample size $n$ because of the independence of $\eta$ on $n$.

**The limitation of the framework in Xiao et al. (2022b)** We would like to note that when the standard loss $f$ satisfies the Assumption 4.1 in Xiao et al. (2022b) (or condition (5) in this paper), in fact every $J-$loss (for any arbitrary $J$, including but not limited to $J^*$) is $2\Gamma_{\mathcal{X}}\epsilon-$approximately smooth. To see this:

$$\begin{aligned} &\|\nabla_{w_1} f_J(w_1, x, y) - \nabla_{w_2} f_J(w_2, x, y)\| \\ =&\|\nabla_{w_1} f(w_1, J(x; y, w_1), y) - \nabla_{w_2} f(w_2, J(x; y, w_2), y)\| & (86) \\ \leq&\beta\|w_1 - w_2\| + \Gamma_{\mathcal{X}}\|J(x; y, w_1) - J(x; y, w_2)\| & (87) \\ \leq&\beta\|w_1 - w_2\| + \Gamma_{\mathcal{X}}(\|J(x; y, w_1) - x\| + \|x - J(x; y, w_2)\|) & (88) \\ \leq&\beta\|w_1 - w_2\| + 2\Gamma_{\mathcal{X}}\epsilon & (89) \end{aligned}$$

where inequality (87) follows from Assumption 4.1 in Xiao et al. (2022b). Inequality (88) and (89) are derived by using the triangle inequality and the condition that $\|J(x; y, w) - x\| \leq \epsilon$ for any $w \in \mathcal{W}$.

Due to the fact that all the $J-$losses have the same approximate smoothness parameter $\eta$, the generalization bounds derived for different $J-$loss, based on the framework in Xiao et al. (2022b), will be the same. This type of generalization bound ignores the influence of the perturbations used in AT on generalization and it is therefore unable to explain the experimental observations in Section 5 and 6 of this paper where different choices of perturbations indeed have distinct impact on generalization.

**Difference of our approach from Xiao et al. (2022b)** In this paper, we depart from the approach of Xiao et al. (2022b), which ignores the specific properties of perturbation $J$, and take a different route which considers the impact of $J$ measured via its expansiveness parameter. Our approach allows us to analyze how different perturbations used in AT affect its generalization performance. Our bounds, derived based on the expansiveness parameter, also avoid having the non-vanishing

term (like the first term in Theorem 5.1 of Xiao et al. (2022b)) when the expansiveness parameter is finite. Only in the case when the expansiveness parameter is unbounded, our results are similar to Xiao et al. (2022b), where the generalization bound contains a non-vanishing term.

The UAS parameter of AT characterizes the gap $\|w - w'\|$ where $w = A(S)$ and $w' = A(S')$ are the model parameters produced by the AT algorithm on two nearly identical datasets $S \simeq S'$. Intuitively, the difference between $w$ and $w'$ arises from the single different example in $S$ and $S'$ (where larger training sample size $n$ tends to reduce the probability of using that single different example to update model parameters in AT), and gets "magnified" by the perturbation $J$ along the AT training trajectory. The expansiveness parameter of $J$ in this paper effectively captures this "magnification" factor. Thus, the eventual difference between $w$ and $w'$ depends on not only the sample size $n$ but also the expansiveness parameter of $J$. Then the exploitation of the expansiveness of $J$ brings sample size $n$ into the bounds.

