# OpenReview forum: "Algorithmic Stability Based Generalization Bounds for Adversarial Training"
_ICLR.cc/2025/Conference — ICLR 2025 Poster_

### Official Review · Reviewer_Jg1M · 2024-10-17

**Soundness:** 3
**Presentation:** 3
**Contribution:** 3
**Rating:** 8
**Confidence:** 4

**Summary:**

This paper provides a new stability bound for adversarial training, where the inner maximization perturbation $J$ and the evaluation perturbation $\pi$ can be different. The introduced term expansiveness can partly explain robust overfitting and experiments are conducted to validate the theoretical results.

**Strengths:**

- The new stability theory differs from existing ones in terms of assumptions and the form of bounds. I like the separation of adversarial training perturbation $J$ and the evaluation perturbation $\pi$, which means that the theory in this paper is a more abstract framework and can be applied in many cases.
- The writing is good.

**Weaknesses:**

- It seems that the framework in this paper can not provide a proper description for the $Q(c^*)$ term, we need to calculate it according to the concrete choice of $J, \pi$. However, to make this framework more significant, examples of how to calculate $Q(c^*)$ and how to choose $c^*$ should be added. Note: I mean examples in practice but not examples with overly simple assumptions (such as the assumption on the second moment in lines 293-294 and the assumption in Corollary 5.2 that a term is bounded by $B$ with probability 1). Just like the VC dimension, if we can not calculate the VC dimension of some hypothesis classes, the bound with the VC dimension is meaningless.
- Typo: in line 149, it should be "the label space $\mathcal{Y}$ is finite"
- A minor issue: many equations in this paper are numbered, in fact, the equations that are not used later need not be numbered. For example, equation (2) is not used.
- In lines 87-88, the paper says that "the bound convergence to a constant, this helps explain the robust overfitting phenomenon". In fact, a lower bound of the generalization gap that converges to a constant can explain overfitting. However, an upper bound can not because your bound may not be tight enough.

**Questions:**

In total, I think this is a good paper, but there are some points that can improve this paper. Please refer to the weaknesses part.

---

> ### Author Response · Authors · 2024-11-26
> **Reply to your comments**
>
> Thank you very much for your careful review and your positive feedback!
>
> Regarding your comments:
>
>
> >- It seems that the framework in this paper can not provide a proper description for the $Q(c^*)$ term, we need to calculate it according to the concrete choice of $J, \pi$. However, to make this framework more significant, examples of how to calculate $Q(c^*)$ and how to choose $c^*$ should be added. Note: I mean examples in practice but not examples with overly simple assumptions (such as the assumption on the second moment in lines 293-294 and the assumption in Corollary 5.2 that a term is bounded by $B$ with probability 1). Just like the VC dimension, if we can not calculate the VC dimension of some hypothesis classes, the bound with the VC dimension is meaningless.
>
>
> We agree that it would be nice to have a better characterization of the term $Q(c^*)$. In this paper, the introduction of this term is to bring in a better handle for driving the bound to decay at the rate of $1/n$. It mainly serves as an analytic technique at the moment. Nonetheless we appreciate your suggestion and will look into the possibility of better characterizing the term when more structural assumptions are incorporated in the loss function or in the network structure.
>
>
>
>
> >- Typo: in line 149, it should be "the label space $\mathcal{Y}$ is finite"
>
>
>
> Thank you very much for pointing out this typo. We have fixed the typo.
>
>
>
> >- A minor issue: many equations in this paper are numbered, in fact, the equations that are not used later need not be numbered. For example, equation (2) is not used.
>
>
>
> Thank you very much for pointing this out. For the time being, to avoid causing any inconvenience for the reviewers during the rebuttal phase, we will retain these equation numbers, allowing the reviewers to easily refer to specific equations in the paper. The unused equation numbers will be removed in the final revised version of the paper.
>
> >- In lines 87-88, the paper says that "the bound convergence to a constant, this helps explain the robust overfitting phenomenon". In fact, a lower bound of the generalization gap that converges to a constant can explain overfitting. However, an upper bound can not because your bound may not be tight enough.
>
>
> You are correct and we fully agree. The statement lacks rigor. We have removed the statement in our manuscript.

---

> > ### Comment · Reviewer_Jg1M · 2024-11-27
> >
> > Thank you for your reply, I have no further questions.

---

### Official Review · Reviewer_Fpbq · 2024-10-21

**Soundness:** 3
**Presentation:** 3
**Contribution:** 3
**Rating:** 6
**Confidence:** 4

**Summary:**

This paper studies the algorithmic stability of adversarial training with a focus on how the inaccuracy of PGD attack affects the stability. Theoretical analysis are provided to justify that the sign function in PGD updates can significantly harm the stability, leading to a worse generalization (gap).

**Strengths:**

(1) This paper is clear and easy to understand.

(2) This paper studies the algorithmic stability of adversarial training from an interesting angle of the PGD attack.

(3) Experiments demonstrate that using tanh to replace sign function can improve the generalization performance.

**Weaknesses:**

(1) While the paper considers the algorithmic stability of PGD attack, a missing component is the convergence of PGD attack. Intuitively, if we always use a fixed attack direction, then the algorithmic stability is not largely affected by the attack. However, the attack is not efficient. When using PGD attack, there is supposed to have a trade-off: with more iterations, the stability gets worse, but the attack becomes stronger. If at the test stage the attacker uses a very strong attack, e.g., AA attack, then balancing the attack effectiveness and stability is essential to obtain a better robust testing performance. Could the authors elaborate more from this perspective?

(2) Please highlight the technical challenges for the theoretical contributions in this paper.

(3) Please consider using some SOTA methods from RobustBench, e.g., leveraging synthetic data in adv training, to conduct the experiments. While improving the sign function seems to be helpful as illustrated by this paper, there is no enough evidence to demonstrate that this is one of the key issues in adversarial training.

(4) Minor: In one line of research, to save the computation budget of adversarial training, algorithms have been proposed to explore fast adversarial training: instead of calculating the attack at each iteration, they update the attack for each sample for one step at each iteration, e.g.,

Cheng, Xiwei, Kexin Fu, and Farzan Farnia. "Stability and Generalization in Free Adversarial Training." arXiv preprint arXiv:2404.08980 (2024).

I'm wondering if the authors can provide some comments on algorithms of this type.

**Questions:**

Please address my comments above.

---

> ### Author Response · Authors · 2024-11-23
> **Reply to your questions and concerns**
>
> Thank you very much for your careful review! To address your concerns and questions, we have performed additional theoretical analysis and experiments and include the results temporarily in Appendix E.2 (from line 1124) of our paper.
>
> Regarding your questions and concerns:
>
> >(1) While the paper considers the algorithmic stability of PGD attack, a missing component is the convergence of PGD attack. Intuitively, if we always use a fixed attack direction, then the algorithmic stability is not largely affected by the attack. However, the attack is not efficient. When using PGD attack, there is supposed to have a trade-off: with more iterations, the stability gets worse, but the attack becomes stronger. If at the test stage the attacker uses a very strong attack, e.g., AA attack, then balancing the attack effectiveness and stability is essential to obtain a better robust testing performance. Could the authors elaborate more from this perspective?
>
> Thank you very much for this interesting question and perspective. We now have included a new section Appendix E.2 in the revised paper, which provides a convergence analysis of the PGD attacks. We now summarize the results here (for more details and more precise statements of our results, please refer to Appendix E.2)
>
> We consider PGD attacks with the mapping $G$ that satisfies the following condition: $\nabla_{x}f(w,x,y)^{T}G(\nabla_{x}f(w,x,y))>0$, for any $(x,y)\in {\cal X}\times {\cal Y}$ and any $w\in{\cal W}$. Note that this condition simply requires the direction of the modified gradient $G(\nabla_{x}f(w,x,y))$ aligned near the direction of the original gradient, within 90 degree angle. Then we have the following result (Lemma E.1 in the revised paper)
>
> **Lemma E.1**  Suppose that $f(w, x, y)$ satisfies the condition (22).Let $ x^*= J^*  (x;y,w)$ and suppose that $\nabla_x f(w, x^*,y)=0$. Suppose $\|G(\nabla_{x}f(w,x,y))\|^2\le C$ for any $(w,x,y)$.  Then performing the $K-$step PGD with step size $\lambda=\frac{1}{\sqrt{K}}$ results in
>
> $ f(w, x^*, y)-\frac{1}{K}\sum_{k=1}^{K}f(w, x^k, y)\le \frac{(2C+d^*)}{2K} + \frac{d^* (\eta^2 + \eta + 1)}{2} $
>
> where $d^* = \max\limits_{k\in \{ 1,\cdots, K \} } \Vert x^k -x^* \Vert ^2$ and $x^k := T^k _{x, y} (x; w)$ denotes the perturbed instance generated by the $k-$step PGD with $k\le K$.
>
> This result bounds the difference between the maximal loss $f(w, x^*, y)$ and the average of the losses achieved by $K$-step PGD (averaged over the $K$ steps). If the achieved loss $f(w, x^k, y)$ increases over the K steps, the result implies
>
>   $  f(w, x^*, y)-f(w, x^K , y)\le \frac{(2C+d^*)}{2K} + \frac{d^* (\eta^2 + \eta + 1)}{2} $
>
>
> Notably this upper bound decays with $K$, but converges to a positive constant. This should come as no surprise since without stronger conditions or knowledge on $f$ (e.g., concavity), it is hopeless to have PGD attacks to reach the true maximal loss value  $f(w,x^*, y)$.
>
> If we further assume loss functions $f(w,x,y)$ to be concave in $x$ and consider the "raw-gradient (RG)"-PGD where the mapping $G$ is taken as the identity map, we have the following convergence upper bound for PGD by directly adapting the Theorem 3.7 in [Bubeck et al, 2015convex] (Lemma E.2 of the revised paper):
>
> **Lemma E.2**  Suppose that $f(w, x, y)$ satisfies the condition (22) and is concave in $x$. Let the mapping $G$ be the identity map. Then the $K-$ step PGD with step size $\lambda=\frac{1}{\eta}$ satisfies
>
>    $ f(w, x^*, y) - f(w, x^K, y) \le \frac{3\eta \|x - x^*\|^2 + f(w, x^*, y)-f(w, x, y)}{K} $
>
> where $x^*= J^* (x;y,w)$ and $x^K:=T^K _{x, y}(x; w)$.
>
> The bound obviously vanishes with $K$.

---

> > ### Author Response · Authors · 2024-11-23
> >
> > **Trade-off between robustness and generalization** We now discuss the tradeoff between robustness and generalization as was brought up in your comments.
> >
> > We will rewrite K-step PGD perturbation as
> >
> >   $  \pi^{\rm PGD} _{K}(x; y, w):= T^K _{x, y}(x; w) $
> >
> > to emphasize its dependency on $K$ in the PGD attack and define the expected robustness gap (on training set)  as
> >
> >   $  {\rm RG}(J^*, \pi):=\mathbb{E} _ {S,A} \left[R_{S}[A_{\pi}(S), J^*] - R_{S}[A_{\pi}(S), \pi] \right] $
> >
> > This term characterizes the robustness of a model on the training set against $J^*$ when it is trained by AT using some other adversarial perturbation $\pi$.
> >
> > For shorter notation, let $w=A_{\pi}(S)$ and consider ${\rm RG} (J^*, \pi^ {\rm PGD} _ {K})$. We can show that
> >
> > ${\rm RG} (J^*, \pi^ {\rm PGD} _ {K})  \le \sup\limits _{(x,y,w)} \left[ f(w, x^*, y)- f(w, x^K, y) \right] $
> >
> > where $x^*= J^* (x;y,w)$ and $x^K :=\pi^{\rm PGD} _{K}(x ;y, w)$.
> >
> > This result and the result above (Lemma E.1 of our revised paper) apply to arbitrary choice of $(w,x,y)$. They suggest that a smaller robustness gap ${\rm RG}(J^*, \pi^ {\rm PGD} _ {K})$ can be achieved for $\pi^{\rm PGD} _ {K}$ with larger $K$. Lemma 5.1 on the other hand suggests that $\pi^ {\rm PGD} _ {K}$ with smaller $K$ tends to achieve a smaller expansiveness parameter $q_{c}(\pi^ {\rm PGD} _ {K})$ and therefore the corresponding generalization gap ${\rm GG} _ {n}(J^*, A_{\pi})$ with $\pi=\pi^ {\rm PGD} _ {K}$ tends to be smaller for smaller $K$.
> >
> > In summary, this theoretical analysis characterizes the potential trade-off between generalization and the "effectiveness of PGD attack" (measured by ${\rm RG}(J^{*}, \pi^{\rm PGD}_{K})$) as was brought up in your comments -- We thank you for this pointer, which has helped improve this paper.

---

> > > ### Author Response · Authors · 2024-11-23
> > >
> > > Regarding your concerns:
> > > >(2) Please highlight the technical challenges for the theoretical contributions in this paper.
> > >
> > > A key challenge in this development lied in identifying the impact of the perturbation operators on the generalization of adversarially trained models. Extensive experiments had been conducted before we came to the recognition that the perturbation operator may play different roles in defining the loss function for evaluation and in the training process and should be isolated for theoretical analysis. Then the stability framework appeared to be a natural option for our analysis. But it remained difficult to find an appropriate measure to characterize the property of the perturbation operator suitable for this analysis. It took a number of iterations before we were able to find an appropriate notion of expansiveness for the perturbation operators.
> > >
> > > >(3) Please consider using some SOTA methods from RobustBench, e.g., leveraging synthetic data in adv training, to conduct the experiments. While improving the sign function seems to be helpful as illustrated by this paper, there is no enough evidence to demonstrate that this is one of the key issues in adversarial training.
> > >
> > >
> > > We have conducted additional experiments on the CIFAR-10 dataset following the AT framework in [Wang et al, 2023] where the model is trained to minimize the TRADES loss proposed in [Zhang et al, 2019] and an additional 1M synthetic dataset is used in the training. Detailed description and results can be found in Appendix E.2 of the revised paper (under "TRADES")
> > >
> > > We conduct experiments to observe if replacing the sign function with the ${\tanh} _ {\gamma}$ function would affect the generalization performance of TRADES. We follow the same setup and hyper-parameter settings in [Wang et al, 2023] and perform TRADES with $G={\tanh} _ {\gamma}$ for $\gamma= 1,10,100,10^3, 10^5 $. Specially, we call this type of TRADES as the ${\rm tanh} _ {\gamma}-$TRADES.
> > >
> > > Models in each experiments are trained for 200 epochs. The trained models are then evaluated by the $J$-(0-1) loss with $J$ taken from  ${\tanh} _ {\gamma}$-PGD, sign-PGD, $J ^{\rm id}$.
> > >
> > > Experimental results are presented in Figure 9(a) where a phenomenon similar to that in Figure 2(a) is observed. When model is trained by ${\tanh}_{\gamma}$-TRADES with smaller $\gamma$, reduced generalization gaps are observed (indicated by the reduced gaps between the dot-shaped and star-shaped curves within each color category).
> > >
> > > Comparing Figure 2(a) with Figure 9(a), one may notice that for larger $\gamma$, the generalization gaps of ${\rm tanh} _ {\gamma}$-TRADES appears to be smaller than those of ${\rm tanh} _ {\gamma}$-PGD-AT. This difference is likely due to the additional 1M synthetic data used in ${\rm tanh} _ {\gamma}$-TRADES while our PGD-AT experiments only utilize the original training dataset which contains far less number of training examples.
> > >
> > > We have also measured the $J$-(0-1) loss with $J$ taken as the ${\rm tanh} _ {\gamma}$-PGD along the training trajectories of ${\rm tanh} _ {\gamma}$-TRADES on both the training and the testing sets. The results, shown in Figure 9(b), use different colors to distinguish ${\rm tanh} _ {\gamma}$-TRADES with different $\gamma$ values. Solid and dashed curves respectively represent the $J$-(0-1) loss on the training and the testing set. It shows that the solid curves drops faster than the dashed curves, indicating that  $J$-(0-1) loss decreases more rapidly for the ${\rm tanh} _ {\gamma}$-TRADES with smaller $\gamma$.
> > >
> > >
> > > In summary, the experimental results indicate that, similar to PGD-AT, the choice of perturbation operators in TRADES also affects its training and generalization performance. On the other hand, we also note that the current analysis in this paper does not fully address the impact of the SIGN function in other adversarial training frameworks, particularly those involving delicate regularization terms, such TRADES. The key difference between TRADES and our set up is in the form of perturbation: our set up restricts the perturbation to a transformation of the gradient of the standard loss, whereas in TRADES alike approaches, the perturbation is a transformation of the gradient of other quantities. Nevertheless, we expect that the general methodology presented in this paper can be adapted to broader families of adversarial training frameworks. -- We sincerely thank the reviewer for bringing up this question, and we will make an effort in that direction.

---

> > > > ### Author Response · Authors · 2024-11-23
> > > >
> > > > Regarding your last question:
> > > >
> > > > >(4) Minor: In one line of research, to save the computation budget of adversarial training, algorithms have been proposed to explore fast adversarial training: instead of calculating the attack at each iteration, they update the attack for each sample for one step at each iteration, e.g.,
> > > > Cheng, Xiwei, Kexin Fu, and Farzan Farnia. "Stability and Generalization in Free Adversarial Training." arXiv preprint arXiv:2404.08980 (2024).
> > > > I'm wondering if the authors can provide some comments on algorithms of this type.
> > > >
> > > >
> > > > Thanks for pointing out this paper to us. The paper studies the generalization performance of free adversarial training and the fast adversarial training (AT). Their theoretical analysis and empirical results suggest that Free AT and Fast AT allows a better generalization performance compared with the vanilla PGD-AT.
> > > >
> > > > Under our framework and with the notations in our paper, Fast AT can be treated as $A_{ \pi}$ with $\pi$ taken as the one-step PGD. As suggested in our Lemma 5.1, PGD with less number of steps $K$ tends to have a smaller expansiveness parameter and the corresponding AT tends to achieve a smaller generalization gap.  Our theory therefore also supports the conclusion in that paper, namely, that Fast AT tends to achieve a better generalization performance.
> > > >
> > > > Free AT, however, has a quite different dynamics from the vanilla PGD-AT discussed in our paper.  Specifically, Free AT may be regarded as a modification of Fast AT, where 1-step PGD does not start from the orignial example $x$, but starts from the previous perturbed version $x^{\rm adv}$ of $x$, until $x$ has undergone $K$ steps of perturbation. This dynamics does not fit immediately to our framework. However, a similar approach as we propose here may be adapted to analyzing Free AT.

---

> ### Author Response · Authors · 2024-12-01
>
> Dear Reviewer,
>
> As the rebuttal period is ending, we hope to hear your comments on our reply to your review. In case you have further questions, please let us know in your earliest convenience, so that we can make an effort to respond before the rebuttal period ends. If we have addressed all your concerns, please consider raising your score. Thank you.

---

### Official Review · Reviewer_2VnC · 2024-10-22

**Soundness:** 2
**Presentation:** 2
**Contribution:** 2
**Rating:** 5
**Confidence:** 3

**Summary:**

This paper studies generalization bounds for robust training by leveraging the framework of uniform stability. The authors analyze $\ell_\infty$ perturbations and derive several upper bounds on the generalization gap of predictors. They then investigate experimentally the performance of adversarially trained models using several algorithms to solve the inner maximization problem of the robust objective.

**Strengths:**

The paper studies an interesting problem: the large generalization gap of robust empirical risk minimization (adversarial training) in neural networks. This work leverages the framework of uniform stability, which has been rather unexplored in the robust learning community, and could potentially provide insights on this topic. Based on the theoretical analyses, the authors propose a sensible relaxation of the commonly used PGD attack, using the $tanh$ function instead. Finally, I agree with the authors that the optimization algorithm in the inner maximization problem has not received adequate attention in the literature, and thus, its study is welcome (despite its limitations—see below).

**Weaknesses:**

The paper is unfortunately difficult to follow, making it challenging to assess its content due to presentation issues. Furthermore, the conclusions seem unoriginal to me. In particular, I identified the following weaknesses:

- Poor presentation: There are many instances where the text is not polished, with numerous grammatical errors (see the non-comprehensive list at the end of the Weaknesses). Additionally, the presentation of the technical results could be substantially improved (e.g., Theorem 4.1: remind the reader of the constants $\beta, \Gamma_X$). Furthermore, the authors should mention in the introduction that all of their results are solely about $\ell_\infty$ perturbations.
- Introduction of many ad-hoc terms to denote well-established concepts: In many places, the authors use obscure words to define concepts that are well-defined in learning theory. For instance, lines 258-259: "mis-matched generalization gap" — this is just the standard generalization gap of a predictor trained robustly. Several such choices make it difficult for readers to comprehend the contributions of this work. Similarly, with so-called "RG-PGD" and the "expansiveness property" (a relaxed notion of Lipschitz continuity).
- Unclear contributions: The paper does not clearly describe how the derived bounds differ from those of Xiao et al. (2022b) and Wang et al. (2024). In particular, the bounds from these prior works are not presented, and they are solely critiqued on the basis that they do not vanish with increasing sample size. Furthermore, the criticism of the non-smoothness of the loss function adopted in prior work seems unfounded ("The source of the non-smoothness is, however, not explained in their work"). Even for linear models under $\ell_\infty$ perturbations, a cross-entropy loss is non-smooth. Hence, the property of non-smoothness is well-motivated.
- Unclear motivation for experiments: The authors seem to identify the sign function in the solution of the inner maximization problem in the robust objective as problematic, and they suggest an alternative based on a smooth approximation. However, they do not show any benefits in terms of robustness with the new method. Furthermore, the fact that for small $\gamma$ we do not observe overfitting and the generalization gap is small appears to be a trivial observation, as the evaluation basically approaches the standard case of no perturbations. In short, it is not a good method for finding worst-case $\ell_\infty$ perturbations.
- Results of Section 6: The authors mention the connection between adversarial training and steepest descent methods, but it is clear that this has been the motivation for the iterative solution of the inner maximization problem since the introduction of adversarial training. Furthermore, the experiments fail to highlight anything new, in my understanding (basically optimising an $\ell_\infty$ objective yields better coverage against $\ell_\infty$ attacks).

Grammatical errors (non comprehensive list):
- in the abstract: "These expansiveness parameters appear not only govern the vanishing rate of the generalization error but also govern its scaling constant."
- line 190: "perturnation" -> perturbation
- line 202: "related with" -> related to
- line 241: "draw" -> draws
- line 245, 256: "descend" -> descent
- line 316: "independent with" -> independent of
- lines 536-537: "Like all up-bound based theoretical results, such an approach is adequate for understanding performance guarantees but may be inadequte to explain poor generalization."

**Questions:**

- Which part of the analysis bypasses prior work and makes the bounds decay as $O(\frac{1}{n})$?
- In the experiments of Section 6, why do you change the threat model (finding different values of $\lambda_p$)? One could imagine experiments with different steepest descent algorithms for the solution of the inner maximization problem, where the threat model does not change (i.e., projecting every time to the same $\ell_\infty$ balls around the original points). Of course, different steepest ascent algorithms (besides the commonly used sign gradient ascent) will perform worse in finding adversarial examples, so the number of inner iterations should be adjusted appropriately. However, I believe this could be an interesting experiment to conduct.

---

> ### Author Response · Authors · 2024-11-26
> **Reply to your concerns and questions (part 1)**
>
> Thank you for taking the time to read our paper. We acknowledge that our manuscript contains minor typos and has room for improvement. Given the extensive use of mathematical notations, we understand that fully grasping the content may require some patience.
>
> However, we would like to note that the other three reviewers have found our paper to be "clear", "easy to follow" and "easy to understand", or "the writing is good".  We have now fixed the typos and kindly invite you to revisit the paper.
>
> For your comments and questions:
>
> >- Poor presentation: There are many instances where the text is not polished, with numerous grammatical errors (see the non-comprehensive list at the end of the Weaknesses). Additionally, the presentation of the technical results could be substantially improved (e.g., Theorem 4.1: remind the reader of the constants $\beta, \Gamma_X$). Furthermore, the authors should mention in the introduction that all of their results are solely about $\ell_\infty$ perturbations.
>
> We have fixed the typos and the grammatical errors that you mentioned.
>
> Regarding your next critique that " the presentation of the technical results could be substantially improved (e.g., Theorem 4.1: remind the reader of the constants $\beta, \Gamma_X$).",  we would like to note that in Theorem 4.1 of our original manuscript, we have explicitly stated at the beginning that "Suppose that $f$ satisfies the conditions (6) and (7). ".   The constants  $\beta, \Gamma_X$  are introduced in conditions (6) and (7).
>
>
>
> Regarding your suggestion that "the authors should mention in the introduction that all of their results are solely about $\ell_\infty$ perturbations.", this has been clearly stated in the original version of the paper (Section 3, current line number 152), as stated "Each adversarial attack (or adversarial perturbation) on input $x$ is assumed to live in an $\infty$-norm ball "
>
> We will consider emphasizing this in the Introduction section of the final manuscript, provided that page limit allows.
>
> >- Introduction of many ad-hoc terms to denote well-established concepts: In many places, the authors use obscure words to define concepts that are well-defined in learning theory. For instance, lines 258-259: "mis-matched generalization gap" — this is just the standard generalization gap of a predictor trained robustly. Several such choices make it difficult for readers to comprehend the contributions of this work. Similarly, with so-called "RG-PGD" and the "expansiveness property" (a relaxed notion of Lipschitz continuity).
>
>
>
> Regarding your critique that "For instance, lines 258-259: 'mis-matched generalization gap' — this is just the standard generalization gap of a predictor trained robustly.",  we suspect that the reviewer might have mis-read this line of statement: this notion is not the standard generalization gap for a predictor trained robustly, but includes it as a special case. Specifically, the "standard generalization gap of a predictor trained robustly" is a type of "mis-matched generalization gap" when $J=J^{\rm id}$ and $\pi$ is a particular perturbation.
>
> We acknowledge that our paper introduces several new terms for the ease of reference and improved clarity. They seem to be have been appreciated by other reviewers.

---

> > ### Comment · Reviewer_2VnC · 2024-11-30
> >
> > Thank you for your response. I will try to be to the point and reply to your answers:
> >
> > >  we would like to note that in Theorem 4.1 of our original manuscript, we have explicitly stated at the beginning that "Suppose that $f$ satisfies the conditions (6) and (7). ". The constants $\beta, \Gamma_X$ are introduced in conditions (6) and (7).
> >
> > This is a matter of style, but I consider it good practice to remind the reader of previously defined quantities in the statement of a theorem (rather than only directing them to previous definitions). This is not crucial.
> >
> > > we suspect that the reviewer might have mis-read this line of statement: this notion is not the standard generalization gap for a predictor trained robustly, but includes it as a special case. Specifically, the "standard generalization gap of a predictor trained robustly" is a type of "mis-matched generalization gap" when $J=J^{\rm id}$ and $\pi$ is a particular perturbation.
> >
> > Right. In line 258, you specifically talk about the case where $J=J^{\rm id}$ and $\pi$ is a particular perturbation. The point is that you could simply call this quantity as "standard generalization gap of a predictor trained robustly", instead of "mis-matched generalization gap". Again, this is not crucial.

---

> ### Author Response · Authors · 2024-11-27
> **Reply to your concerns and questions (part 2)**
>
> >- Unclear contributions: The paper does not clearly describe how the derived bounds differ from those of Xiao et al. (2022b) and Wang et al. (2024). In particular, the bounds from these prior works are not presented, and they are solely critiqued on the basis that they do not vanish with increasing sample size. Furthermore, the criticism of the non-smoothness of the loss function adopted in prior work seems unfounded ("The source of the non-smoothness is, however, not explained in their work"). Even for linear models under $\ell_\infty$ perturbations, a cross-entropy loss is non-smooth. Hence, the property of non-smoothness is well-motivated.
>
>
>
> Regarding your comment that "The paper does not clearly describe how the derived bounds differ from those of Xiao et al. (2022b) and Wang et al. (2024)...",  we note that this has been clarified in our response to Reviewer e9T9 and we also have incorporated the detailed comparison with Xiao et al. (2022b) and Wang et al. (2024) into the revised manuscript from Line 1392 in Appendix F. We invite the reviewer to read that discussion.
>
> Regarding your next critique on our statement that "The source of the non-smoothness is, however, not explained in their work", we believe there might have been some misinterpretation. To clarify, this statement is not intended as a critique of Xing et al. (2021) for assuming non-smoothness of the adversarial loss. Rather, it highlights that the non-smoothness property used in Xing et al (2021) is not characterized at any quantitative level. In fact, the development of Xing et al (2021) does not rely on any quantitative specification of the non-smoothness and directly invokes the previous result of Bassily et al (2020) .
>
> Nonetheless we recognize that this statement wasn't clear enough, and we have revised it to "The non-smoothness is however not quantitatively characterized in their work".
>
>
>
>
>
>
>
> >- Unclear motivation for experiments: The authors seem to identify the sign function in the solution of the inner maximization problem in the robust objective as problematic, and they suggest an alternative based on a smooth approximation. However, they do not show any benefits in terms of robustness with the new method. Furthermore, the fact that for small $\gamma$ we do not observe overfitting and the generalization gap is small appears to be a trivial observation, as the evaluation basically approaches the standard case of no perturbations. In short, it is not a good method for finding worst-case $\ell_\infty$ perturbations.
>
>
> We suspect that the reviewer misunderstood this part of the paper.
>
> The motivation of introducing ${\rm tanh}_{\gamma}$ function in our experiments stems directly from the theoretical analysis presented in the previous section, where the purpose, as we stated in the original paper (line 387 in the current manuscript), is "To investigate how the expansiveness property affects generalization". It is not proposing a new AT algorithm.  The experiments are designed to empirically validate our theoretical analysis.

---

> > ### Comment · Reviewer_2VnC · 2024-11-30
> >
> > > we note that this has been clarified in our response to Reviewer e9T9 and we also have incorporated the detailed comparison with Xiao et al. (2022b) and Wang et al. (2024) into the revised manuscript from Line 1392 in Appendix F. We invite the reviewer to read that discussion.
> >
> > The included discussion seems to be good. I also welcome the fact that the comments on the smoothness assumptions of prior works have been rectified.
> >
> > > the purpose, as we stated in the original paper (line 387 in the current manuscript), is "To investigate how the expansiveness property affects generalization". It is not proposing a new AT algorithm. The experiments are designed to empirically validate our theoretical analysis.
> >
> > Indeed. My critique of "However, they do not show any benefits in terms of robustness with the new method." was not relevant. However, the rest of my critique still holds as far as I see:  "Furthermore, the fact that [...] for finding worst-case  perturbations."

---

> ### Author Response · Authors · 2024-11-27
> **Reply to your concerns and questions (part 3)**
>
> >- Results of Section 6: The authors mention the connection between adversarial training and steepest descent methods, but it is clear that this has been the motivation for the iterative solution of the inner maximization problem since the introduction of adversarial training. Furthermore, the experiments fail to highlight anything new, in my understanding (basically optimising an $\ell_\infty$ objective yields better coverage against $\ell_\infty$ attacks).
>
>
>
> There have been a misunderstanding regarding Section 6 of our paper. This section is not intended to explain "the connection between adversarial training and steepest descent methods," as stated by the reviewer. The beginning of the section intends to explain why the sign function (or the sign gradient) is specifically used in PGD rather than the raw gradient. Note that this peculiar choice of sign function is not related to the $\infty-$norm ball used in the projection step of PGD. We will elaborate this below.
>
> In Section 6, we investigate the effects of replacing the sign function with alternative operators $G_p$. Importantly, the projection step in PGD consistently operates with respect to the same $\infty-$norm ball, regardless of the choice of $G_p$.
>
>
>
> Each iteration of PGD involves a gradient ascent step followed by a projection step. The emergence of the sign function arises from treating the **gradient ascent step** as maximizing a locally linear approximation of the loss function.
>
>
>
> Specifically, at the $k-$th iteration of PGD where  $x^ k$ is to be updated by PGD,  the loss function is approximately considered to be linear within a $p-$norm ball (i.e., $ \mathbb{B} _ p (x^ k , \lambda _ p) $ ) around $x^ k$ (not around $x$) with radius $\lambda _ p$ . If this norm ball is chosen as $\mathbb{B} _ {\infty}(x^ k, \lambda)$, the sign function naturally arises in the gradient ascend step of the PGD. However, this does not mean choosing $\mathbb{B} _ p (x^ k, \lambda _ p)$ as $\mathbb{B} _ {\infty}(x^ k, \lambda)$ is the only option. We therefore investigate the norm ball $\mathbb{B}_{p}(x^k, \lambda_p)$ with other choices of $p$. This merely corresponds to the linear approximation of the loss function within different local ranges. Consequently, $G_p$ in different forms results.
>
> It is also important to emphasize that both $\mathbb{B} _ {p}(x^ k, \lambda _ p)$ and $\mathbb{B} _ {\infty}(x^ k, \lambda)$ are different from $\mathbb{B} _ {\infty}(x, \epsilon)$ that is used in the projection step of PGD. Regardless the choice of  $\mathbb{B} _ {p}(x^ k, \lambda _ p)$ in determing  $G _ p$ for the gradient ascend step of PGD, the projection step always operates using the same $\infty-$norm ball $\mathbb{B}_{\infty}(x, \epsilon)$ . We suspect the reviewer might have confused these two types of norm balls when first reading our paper, leading to a misunderstanding of Section 6.
>
>
> Finally, to the best of our knowledge, the perspective of replacing the sign function with $G_p$ and analyzing its effects is novel. Contrary to the comment that this approach "fails to highlight anything new," we believe this original angle offers valuable insights into understanding the generalization of AT.

---

> > ### Author Response · Authors · 2024-11-27
> > **Reply to your concerns and questions (part 4)**
> >
> > Regarding your questions:
> >
> > >- Which part of the analysis bypasses prior work and makes the bounds decay as $O(\frac{1}{n})$?
> >
> >
> > The work in  Xiao et al. (2022b) define the notion of "$\eta-$approximate smoothness"  for loss functions and derive generalization bounds based on this quantity. The subsequent work by Wang et al. (2024) builds upon this framework. We believe that  the primary reason their bounds include terms that do not vanish with $n$ is that the approximate smoothness paramete $\eta$ is independent of $n$. A more detailed discussion of their framework is provided in Appendix F of our revised manuscript.
> >
> >
> >
> > Our work, on the other hand, derives generalization bounds from a different perspective. We define the notion of  $c-$expansiveness for the perturbation operations used in AT. Our bounds derived based on this quantity then address the limitations in  Xiao et al. (2022b) and Wang et al. (2024), and vanish with $O(\frac{1}{n})$ when the expansiveness parameter is finite.
> >
> >
> >
> >
> >
> > >- In the experiments of Section 6, why do you change the threat model (finding different values of $\lambda_p$)? One could imagine experiments with different steepest descent algorithms for the solution of the inner maximization problem, where the threat model does not change (i.e., projecting every time to the same $\ell_\infty$ balls around the original points). Of course, different steepest ascent algorithms (besides the commonly used sign gradient ascent) will perform worse in finding adversarial examples, so the number of inner iterations should be adjusted appropriately. However, I believe this could be an interesting experiment to conduct.
> >
> >
> >
> > As explained above, the threat model (i.e., $\mathbb{B} _ {\infty}(x, \epsilon)$) remains unchanged in the experiments of Section 6. What varies is $\mathbb{B} _ {p}(x ^k, \lambda _ p)$, which is used to determine $G_p$ for the gradient ascent step of PGD. Finding different values of $\lambda_p$ is to maintain the same volume for the balls $\mathbb{B} _{p}(x ^k,\lambda _ p)$ across different $p$ values.

---

> > > ### Comment · Reviewer_2VnC · 2024-11-30
> > >
> > > > the threat model [...] remains unchanged in the experiments of Section 6.
> > >
> > > I see. Thanks for the clarification. I mistakenly thought that you changed the threat model, instead of the step size. The experiments now make more sense, and basically do what I originally asked for. However, a question remains about the number of inner iterations: how did you change this value when changing the inner steepest ascent method? Did you vary it at all?

---

> > > > ### Author Response · Authors · 2024-12-01
> > > > **Reply to your follow-up questions and your remaining concerns (part 1)**
> > > >
> > > > Thank you very much for taking the time to revisit our paper. We address your further comments as below.
> > > >
> > > >
> > > > >- This is a matter of style, but I consider it good practice to remind the reader of previously defined quantities in the statement of a theorem (rather than only directing them to previous definitions). This is not crucial.
> > > >
> > > > >- Right. In line 258, you specifically talk about the case where $J=J^{\rm id}$ and $\pi$ is a particular perturbation. The point is that you could simply call this quantity as "standard generalization gap of a predictor trained robustly", instead of "mis-matched generalization gap". Again, this is not crucial.
> > > >
> > > >
> > > > Thank you for the suggestion and we will consider it during next revision -- currently we are not allowed to revise the manuscript.
> > > >
> > > >
> > > > >- Indeed. My critique of "However, they do not show any benefits in terms of robustness with the new method." was not relevant. However, the rest of my critique still holds as far as I see: "Furthermore, the fact that for small $\gamma$ we do not observe overfitting and the generalization gap is small appears to be a trivial observation, as the evaluation basically approaches the standard case of no perturbations. In short, it is not a good method for finding worst-case $\ell_\infty$ perturbations.
> > > >
> > > >
> > > >  When $\gamma$ is zero or close to zero, we agree with your comment that it "basically approaches the standard case of no perturbations." However, beyond such an extreme case, for example when $\gamma$ takes a relatively large value (e.g., $\gamma=10, 10^2, 10^3$), the relationship between robust generalization and the strength of the perturbation (or its optimality with respect to achieving the worst-case $\ell_\infty$ perturbation) used in AT is not clear.
> > > >
> > > > For instance, as demonstrated in Figure 1, adversarial training (AT) using three-step sign-PGD is far from optimal for finding the worst-case perturbations, yet it results in robust overfitting.  On the other hand, ${\tanh}_{\gamma}$-PGD with small $\gamma$ is similarly non-optimal for finding the worst-case perturbations, it  however enables AT to achieve better generalization performance.
> > > >
> > > > Therefore, the generalization performance for ${\tanh}_{\gamma}$-PGD-AT in relation to the value of $\gamma$ is not automatically clear without theoretical study or experimental investigation. As such, our experimental results are not trivial.

---

> > > > > ### Author Response · Authors · 2024-12-01
> > > > > **Reply to your follow-up questions and your remaining concerns (part 2)**
> > > > >
> > > > > >- No, I think I understood what you wrote in the paper in the first place. Perhaps my comment about steepest descent has not been clear enough. I understand that that Section is about the update rule, and not the projection step. Maximizing the linear approximation of the loss around the current iterate around a ball induced by a norm $|\cdot|$ is equivalent to using steepest descent (ascent) on the loss with respect to this norm -- see, for instance, Section 9.4 in 'Convex Optimization by Boyd and Vandenberghe'.
> > > > >
> > > > > We are glad that your concern did not arise from the confusion between the norm ball used in gradient asecent and that used in the projection operation. Now returning to your original comment "the experiments fail to highlight anything new", we remark the following.
> > > > >
> > > > >
> > > > > From Sections 5, we see that the gradient operator $G$ has a significant impact on generalization. In Section 6, we recognized that sign operator, as the gradient operator, results from solving the inner maximization problem by a locally linear approximation of the loss function. Changing the range of locally linear approximation from $\ell_\infty$ ball to other norm balls, results in a family of gradient operators, i.e., the operators $G_p$'s. The results in this section are NEW in three aspects:
> > > > > 1.  We observe that the impact of the operator $G_p$ on generalization of AT, depends on the value of $p$.
> > > > > 2.  We present a theoretical explanation of the above observations. Specifically in Lemma 6.1, we relate the Lipschitz constant $\alpha_p$ of the $G_p$ operator to the value of $p$. The Lipschitz constant in turn affects the expansiveness of $G_p$-PGD (as shown in Lemma 5.1), and hence impacts generalization (as suggested by equation (15)).
> > > > > 3.  The family of $G_p$ gradient operators appear to result in poor generalization. This potentially points to a fundamental limitation in the approach to solve inner maximization using locally linear approximation of the loss function.
> > > > >
> > > > > >- However, a question remains about the number of inner iterations: how did you change this value when changing the inner steepest ascent method? Did you vary it at all?
> > > > >
> > > > > We would like to clarify that the number of iterations for each $G_p-$PGD variants remains unchanged. We however have adjusted the value of $\lambda_p$ (the step size of the inner iteration) to ensure that the volumes of $\mathbb{B}_{p}(x^k, \lambda_p)$ are the same across different values of $p$ for a fair comparison.
> > > > >
> > > > >
> > > > > Thank you again for your response. We hope that our clarifications have addressed your remaining concerns. Meanwhile, as several of your earlier concerns are now resolved, we hope you consider raising your score. Should there be other issues for which you need our clarification, please let us know and we will make an effort to explain.

---

> > > > > > ### Comment · Reviewer_2VnC · 2024-12-03
> > > > > >
> > > > > > Thank you for your reply and your clarifications.
> > > > > >
> > > > > > > We would like to clarify that the number of iterations for each $G_p-$PGD variants remains unchanged.
> > > > > >
> > > > > > For future reference, I think an exploration of learning step together with number of iterations would have been interesting.
> > > > > >
> > > > > > I revisited the paper and most of my concerns regarding presentation and contributions have not been met unfortunately. However, I do see some value in the experiments of Section 6 (which I missed in my original evaluation), so I updated my scores accordingly. However, I still recommend (borderline) rejection for most of the points raised in the Weaknesses section of my review.

---

> > > > > > > ### Author Response · Authors · 2024-12-03
> > > > > > >
> > > > > > > Dear reviewer,
> > > > > > >
> > > > > > >
> > > > > > > Thank you for revisiting the paper and raising the score. We are however surprised to see your comment that "most of my concerns regarding presentation and contributions have not been met unfortunately".  Based on our discussions and your responses, we believe that most of your concerns have been effectively addressed, as can be seen from a revisit of your initial comments below:
> > > > > > >
> > > > > > > - "Poor presentation ..."
> > > > > > >
> > > > > > >   In your earlier comments, you noted, *"This is a matter of style... This is not crucial."* where you indicate that this is less critical for your evaluation.
> > > > > > >
> > > > > > > - "Introduction of many ad-hoc terms to denote well-established concepts ..."
> > > > > > >
> > > > > > >   Similarly, you mentioned, *"Again, this is not crucial."* Hence, we interpreted this as a minor stylistic issue rather than a significant drawback.
> > > > > > >
> > > > > > > - "Unclear contributions..."
> > > > > > >
> > > > > > > In response to our earlier clarification, you remarked, *"The included discussion seems to be good. I also welcome the fact that the comments on the smoothness assumptions of prior works have been rectified."* We interpreted this as an acknowledgment that the concern has been addressed.
> > > > > > >
> > > > > > > - "Unclear motivation for experiments ..."
> > > > > > >
> > > > > > > Based on your feedback "Indeed. My critique of 'However, they do not show any benefits in terms of robustness with the new method.' was not relevant.", we suppose this concern has been partially addressed in our first round of clarification.
> > > > > > >
> > > > > > > You then mentioned that "However, the rest of my critique still holds as far as I see....". We have provided further clarifications addressing the remaining concerns and would like to know if there are any aspects still needing elaboration.
> > > > > > >
> > > > > > > - "Results of Section 6..."
> > > > > > >
> > > > > > > In your initial feedback, you mentioned, "Thanks for the clarification... The experiments now make more sense." Furthermore, in your current comment, you acknowledged, "I do see some value in the experiments of Section 6 (which I missed in my original evaluation)." Based on this,  your concerns about Section 6 appear to have been resolved.
> > > > > > >
> > > > > > >
> > > > > > > In summary, from your previous feedback during the discussions, we felt that most of your concerns have been addressed adequately. Now that you suggest this is not the case, we will appreciate that you elaborate on your concerns so that we can make our best effort clarifying.

---

> > ### Comment · Reviewer_2VnC · 2024-11-30
> >
> > No, I think I understood what you wrote in the paper in the first place. Perhaps my comment about steepest descent has not been clear enough. I understand that that Section is about the update rule, and not the projection step. Maximizing the linear approximation of the loss around the current iterate around a ball induced by a norm $\|\cdot\|$ is equivalent to using steepest descent (ascent) on the loss with respect to this norm -- see, for instance, Section 9.4 in "Convex Optimization by Boyd and Vandenberghe".

---

### Official Review · Reviewer_e9T9 · 2024-10-29

**Soundness:** 2
**Presentation:** 3
**Contribution:** 3
**Rating:** 6
**Confidence:** 3

**Summary:**

Authors study the generalization of adversarial training with projected gradient descent. They provide uniform stability upper bounds of the generalization gap that consider the expansiveness of the adversarial attack operator. In the particular case of replacing the $\text{sign}(x)$ operation in the PGD attack with $\text{tanh}(\gamma\cdot x)$, they can show that their generalization upper bound decays to zero with the number of samples $n$ for a finite value of $\gamma$. The experimental evaluation shows the tradeoff between generalization and robustness given by $\gamma$, where smaller values of $gamma$ obtain good generalization but poor robustness and the opposite happens for larger $\gamma$.

**Strengths:**

- Simple theory and easy to follow paper. I didn’t read the proofs in full detail, but the main paper is easy to follow and the analysis and experiments are reasonable.

- I found the analysis of the expansiveness of the adversarial attack operator very interesting and up to my knowledge, this has not been considered before.

- When considering finite $\gamma$, authors can show that their generalization upper bound converges to zero with increasing number of training samples $n$.

**Weaknesses:**

- Authors claim that their upper bound converges to zero with increasing number of training samples $n$ and the ones of [1,2] do not. This is misleading as [1,2] do not consider the expansiveness of the attack operator and the bound provided in this work, in the same setup as [1,2] does not vanish with $n$ (see lines 369-371).

- The difference with the previous bounds is not clearly covered in the paper. The proof technique and assumptions are very similar to [1,2], nevertheless the bounds in [1,2] are not presented in the work and there is no discussion about how to integrate previous bounds with the expansiveness setup introduced in this work, making it difficult to assess the contributions. It would be nice to add a discussion about which role expansiveness plays in the result of [1,2], i.e., can it result in a vanishing upper bound with $n$? It would also be good to have a table comparing the different upper bounds.

**Questions:**

- Some small typos:
	- Line 190: pernutation -> permutation
	- Line 267: exist -> exists
	- Line 483: It then curious -> It is then curious

- How does the bound in [1,2] change when considering $\text{tahn}_{\gamma}$-PGD?
- Have you tried larger $\gamma$s? $\gamma = 10^{5}$ seems to be very far from sign-PGD in Figure 2 (b).  It would be interesting to see how does $\text{tahn}_{\gamma}$-PGD behave when it’s close to sign-PGD.
- Can you construct an experiment where the dependence on $n$ is displayed? For example taking a smaller number of samples from the studied datasets in order to see how the generalization gap grows. A synthetic distribution could also be employed where more and more samples are drawn and the gap decreases to zero for finite $\gamma$.

**References:**

[1] Wang et al., Data-dependent stability analysis of adversarial training, ArXiv 2024.

[2] Xiao et al., Stability Analysis and Generalization Bounds of Adversarial Training, NeurIPS 2022

---

> ### Author Response · Authors · 2024-11-19
> **Reply to your concerns and questions**
>
> Thank you very much for your careful review! We have fixed the typo in our paper. To address your concerns and questions, we have conducted additional experiments and include the results temporarily in Appendix E of our paper.
>
> Regarding your concerns:
>
> - "Authors claim that their upper bound converges to zero with increasing number of training samples and the ones of [1,2] do not. This is misleading as [1,2] do not consider the expansiveness of the attack operator and the bound provided in this work, in the same setup as [1,2] does not vanish with $n$ (see lines 369-371)."
>
> We acknowledge that our bound does not vanish when $J$-loss defined using sign-PGD. However, we note that although the behavior of our bound in this specific setting is similar to that in [1,2], our setup for this result and the setup of the results in [1,2] should not be taken as the same. For example, the main result in [2] (Theorem 5.1) states, using our notations, that for any $J$-loss satisfying the various assumptions in [2], the geneneralization bound does not vanish. However, in our paper, we show that only for $J$-loss defined using sign-PGD, our bound does not vanish.
>
> For $J$-losses satisfying the conditions of Theorem 5.1 in [2], we show that there exists a broad family of $J$-losses for which our generalization bound does vanish. That is, for this family of $J$-losses (namely, those having bounded expansiveness), our results are stronger than that in [1, 2]. This is further elaborated when we address your next comment.
>
> - "The difference with the previous bounds is not clearly covered in the paper. The proof technique and assumptions are very similar to [1,2], nevertheless the bounds in [1,2] are not presented in the work and there is no discussion about how to integrate previous bounds with the expansiveness setup introduced in this work, making it difficult to assess the contributions. It would be nice to add a discussion about which role expansiveness plays in the result of [1,2], i.e., can it result in a vanishing upper bound with $n$?  It would also be good to have a table comparing the different upper bounds."
>
> Since the work of [1] is built upon the framework in [2], we here only present the connections and differences between [2] and our work.
>
> **Summary of generalization bounds in [2]:** First we would like to note that our problem setting includes the setting in [2] as a special case. Specifically, the generalization gap discussed in [2] corresponds to the generalization gap ${ \rm GG} _ {n} (J^*, A _ {J^*})$ defined in our work, where the perturbations in both $J-$loss and the AT algorithm are taken as the optimal adversarial perturbation $J^{*}$.
>
> Our work and [2] both take the Lipschitzness and smoothness conditions of the standard loss $f$ as the starting point, but derive generalization bounds from different perspectives:  the work in [2] defines and proposes to study the $\eta-$approximate smoothness of the adversarial loss ( $f^*$ in our notation) and derive generalization bounds based on this quantity. Our work define the notion of $c-$expansiveness of the perturbation operator (e.g., $J^{*}$) and show how this quantity affects generalization performance of AT.
>
> For completeness, we here present the definition of $\eta-$approximate smoothness, rewrite the Definition 4.1 of [2] using our notations.
>
> **Definition** ($\eta-$approximate smoothness [2])  A loss function $f_J$  is called  $\eta-$approximately $\beta-$gradient Lipschitz if there exists  $\beta>0$ and $\eta>0$ such that for any $(x,y)\in {\cal X}\times {\cal Y}$  and for any $w_1, w_2 \in {\cal W}$ we have
> ​                  $$  \Vert \nabla f_J(w_1, x, y)-\nabla f_J(w_2, x, y)  \Vert \le \beta \Vert w_1-w_2 \Vert + \eta$$
> The work in [2] then derives generalization bounds for loss functions that are $\eta-$approximately smooth.  For example,  after replacing the notations in [2] with ours, Theorem 5.1 of [2] shows that if $f_{J}$ is $\eta-$approximately $\beta-$gradient Lipschitz, convex in $w$ for all $(x,y)$ and the standard loss $f$ satifies the same Lipschitz condition in (6) of our paper (or Assumption 4.1. in [2]), then their bound in Theorem 5.1 becomes
> ​                         $${\rm GG}_ {n} (J, A _ {J}) \le \frac{L _ {\cal W}}{\beta}\eta T + \frac{2L_{\cal W}^{2}}{n\beta}T$$
> The authors of [2] show that the adversarial loss $f^*$ satisties $\eta$-approximately $\beta$-gradient Lipschitz with $\eta = 2\Gamma _{\cal X}\epsilon$ so that the generalization bound above gives their generalization bound for adversarial training. In their determination of the $\eta$ parameter, they have assumed that the standard loss $f$ satisfies certain Lipschitz and smoothness condition; this condition is effectively equivalent to our condition (7).
>
> It is worth noting that the generaliztion bounds derived based on the approximate smoothness parameter $\eta$ contain a term unrelated to the sample size $n$ because of the independence of $\eta$ on $n$.

---

> ### Author Response · Authors · 2024-11-19
>
> **The limitation of the framework in [2]:**  We would like to note that when the standard loss $f$ satisfies the Assumption 4.1 in [2] (or condition (7) in our paper), in fact every $J-$loss (for any arbitrary $J$, including but not limited to $J^*$) is $2\Gamma_{\cal X}\epsilon-$approximately smooth. To see this:
>
> $\Vert \nabla_{w_1}f_{J}(w_1, x, y)-\nabla_{w_2} f_{J}(w_2, x, y)\Vert$
>
> $=\Vert\nabla_{w_1}f(w_1, J(x;y, w_1), y)-\nabla_{w_2}f(w_2, J(x;y, w_2), y)\Vert$
>
> $\le \beta \Vert w_1 - w_2 \Vert + \Gamma_{\cal X} \Vert J(x;y, w_1) - J(x;y, w_2) \Vert \quad (1)$
>
> $\le \beta \Vert w_1 - w_2 \Vert + \Gamma_{\cal X} (\Vert J(x;y, w_1) - x \Vert+ \Vert x-J(x;y, w_2) \Vert)\quad (2)$
>
> $\le \beta \Vert w_1 - w_2 \Vert + 2\Gamma_{\cal X}\epsilon \quad (3)$
>
> where inequality (1) follows from Assumption 4.1 in [2]. Inequality (2) and (3) are derived by using the triangle inequality and the condition that $\Vert J(x;y,w)-x \Vert \le \epsilon$ for any $w \in {\cal W}$.
>
> Due to the fact that all the $J-$losses have the same approximate smoothness parameter $\eta$, the generalization bounds derived for different $J-$loss, based on the framework in [2],  will be the same. This type of generalization bound ignores the influence of the perturbations used in AT on generalization and is therefore unable to explain the experimental observations in our work where different choices of perturbations indeed have distinct impact on generalization.
>
> **Difference of our approach from [2]:** In this paper, we depart from the approach of [2], which ignores the specific properties of perturbation $J$, and take a different route which considers the impact of $J$ measured via its expansiveness parameter. Our approach allows us to analyze how different perturbations used in AT affect its generalization performance. Our bounds, derived based on the expansiveness parameter, also avoid having the non-vanishing term (like the first term in Theorem 5.1 of [2]) when the expansiveness parameter is finite. Only in the case when the expansiveness parameter is unbounded, our results are similar to [2], where the generalization bound contains a non-vanishing term.
>
> The UAS parameter of AT characterizes the gap $\Vert w-w' \Vert$ where $w=A(S)$ and $w'=A(S')$ are the model parameters produced by the AT algorithm on two nearly identical datasets $S\simeq S'$. Intuitively, the difference between $w$ and $w'$ arises from the single different example in $S$ and $S'$ (where larger training sample size $n$ tends to reduce the probability of using that single different example to update model parameters in AT), and gets "magnified" by the perturbation $J$ along the AT training trajectory. The expansiveness parameter of $J$ that we define  effectively captures this "magnification" factor. Thus, the eventual difference between $w$ and $w'$ depends on not only the sample size $n$ but also the expansiveness parameter of $J$. Then our exploitation of the expansiveness of $J$ brings sample size $n$ into our bound.

---

> ### Author Response · Authors · 2024-11-19
>
> Regarding your questions:
>
>
> - "How does the bound in [1,2] change when considering ${\rm tanh}_{\gamma}$-PGD?"
>
> As we have shown above, the bound remains unchanged in [1,2] when considering ${\rm tanh}_{\gamma}$-PGD with different $\gamma$.
>
> - "Have you tried larger $\gamma$ s?  $\gamma=10^5$ seems to be very far from sign-PGD in Figure 2 (b). It would be interesting to see how does tanh-PGD behave when it’s close to sign-PGD."
>
>
> The reviewer might have misread Figure 2b. In fact, in that figure when $\gamma = 10^5$, ${\rm tanh}_{\gamma}$ function closely approximates the sign function and $\gamma = 10^5$ is actually sufficiently large.
>
> To illustrate, we have plotted the training trajectory of ${\rm tanh}_{\gamma}$-PGD AT with $\gamma=10^5$ and the trajectory of the sign-PGD AT in Figure 7 (a), Appendix E.1. (from line 1074). The results show that their trajectories overlap almost entirely.
>
> Further, we have plotted the $\tanh_\gamma$ function with $\gamma=10^5$ and the sign function in Figure 7 (b), Appendix E.1, to show that they are indeed very close.
>
> -  "Can you construct an experiment where the dependence on  $n$ is displayed? For example taking a smaller number of samples from the studied datasets in order to see how the generalization gap grows. A synthetic distribution could also be employed where more and more samples are drawn and the gap decreases to zero for finite $\gamma$."
>
> Upon your request, we have performed AT experiments using various fraction of the training set from CIFAR-10 and SVHN. The generalization gaps are estimated and plotted in Figure 8 in Appendix E.1. It is clear that the generalization gaps reduce with the increase of training sample size $n$. Notably, in the limit when $n$ approaches infinity, the model is effectly trained on the entire data distribution, and the generalization gap must approach zero.
>
> Also observed from the figure is the phenomenon that smaller $\gamma$ gives smaller generalization gap. This is consistent with our theoretical analysis.

---

> > ### Comment · Reviewer_e9T9 · 2024-11-25
> > **Thanks for your response**
> >
> > Dear Authors,
> >
> > I am happy with the clarifications and appreciate your efforts in this rebuttal. I was expecting the analysis regarding the comparison with [1,2] to be included in the revised version. I believe this **must** be included in the manuscript. Overall I am satisfied and have increased my score.

---

> ### Author Response · Authors · 2024-11-26
> **Thank you for raising your score**
>
> Dear reviewer,
>
> Thank you for raising your score! We have now included comparison of our work with [1, 2] in Appendix F. We also would like to elaborate this aspect in the main body of the paper. But at current stage, in order not to disturb line numbers for the ease of all reviewers, we will postpone this when we prepare next version of the paper.

---

### Meta-Review · Area_Chair_AUr2 · 2024-12-13

**Metareview:**

The theory of adversarial training has been substantially explored by many works. But, this paper present a novel stability analysis of adversarial training and prove generalization upper bounds in terms of an expansiveness property of adversarial perturbations. The proof technique used in this paper is totally different from the previous papers. The derived bound is more general than existing works. This is really a good theory job in adversarial training. Congratulations!

**Additional Comments On Reviewer Discussion:**

A reviewer seems to have bias on this paper and ask some quesitons that do not really matter.

---

### Decision · Program_Chairs · 2025-01-22

Accept (Poster)